# Splat Feature Solver

**Butian Xiong[1], Rong Liu[1], Kenneth Xu[2], Meida Chen[1], Andrew Feng[1]**
[1]University of Southern California, Institute for Creative Technologies
[2]University of Michigan, Ann Arbor

## Abstract

Feature lifting has emerged as a crucial component in 3D scene understanding, enabling the attachment of rich image feature descriptors (e.g., DINO, CLIP) onto splat-based 3D representations. The core challenge lies in optimally assigning rich general attributes to 3D primitives while addressing the inconsistency issues from multi-view images. We present a unified, kernel- and feature-agnostic formulation of the feature lifting problem as a sparse linear inverse problem, which can be solved efficiently in closed form. Our approach admits a provable upper bound on the global optimal error under convex losses for delivering high quality lifted features. To address inconsistencies and noise in multi-view observations, we introduce two complementary regularization strategies to stabilize the solution and enhance semantic fidelity. Tikhonov Guidance enforces numerical stability through soft diagonal dominance, while Post-Lifting Aggregation filters noisy inputs via feature clustering. Extensive experiments demonstrate that our approach achieves state-of-the-art performance on open-vocabulary 3D segmentation benchmarks, outperforming training-based, grouping-based, and heuristic-forward baselines while producing lifted features in minutes. Our **code** is available in the GitHub. We provide additional website for more visualization, as well as the video.

## 1 Introduction

Recent advances in splat representations—such as 3D Gaussian Splatting (3DGS) Kerbl et al. (2023), 2D Gaussian Splatting Huang et al. (2024), and Deformable Beta Splatting Liu et al. (2025)—have enabled real-time, high-fidelity scene rendering by modeling geometry with compact, explicit primitives. These splat-based methods combine differentiable projection, alpha compositing, and efficient visibility-aware rasterization to preserve geometric detail and multi-view consistency, supporting applications ranging from style transfer Liu et al. (2024); Galerne et al. (2025) to scene understanding Guo et al. (2024); Qin et al. (2024); Shi et al. (2024); Jun-Seong et al. (2025); Cheng et al. (2024); Cen et al. (2025); Dou et al. (2024); Gu et al. (2024); Zuo et al. (2025); Peng et al. (2024b). Nevertheless, enriching these primitives with detailed descriptors—such as CLIP, DINO, General ViT, and CNN features—remains challenging due to the inherent difficulty of coherently lifting 2D observations into consistent 3D representations.

To address this challenge, recent efforts have focused on lifting per-pixel semantic descriptors (e.g., CLIP, DINO) onto 3D primitives, enabling semantic tasks such as segmentation and querying. Existing semantic feature lifting methods broadly fall into three categories: training-based optimization Shi et al. (2024); Qin et al. (2024); Zhou et al. (2024); Qiu et al. (2024); Zuo et al. (2025), which pioneered in embedding semantic onto 3D primitives via multi-view training; grouping-based association Wu et al. (2025); Peng et al. (2024b); Gu et al. (2024); Liang et al. (2024), which improved efficiency through feature clustering; and heuristic forward methods Guo et al. (2024); Joseph et al. (2024); Dou et al. (2024); Chacko et al. (2025); Jun-Seong et al. (2025); Cheng et al. (2024), which prioritize speed and directly project semantic features onto 3D primitives.

While existing methods have achieved promising results, a unified theoretical framework for feature lifting remains underexplored, especially concerning the heuristic forward methods that bypass the training process. Building upon prior efforts, we identify several open challenges that limit generalization and theoretical understanding. First, establishing a rigorous mathematical formulation for defining feature lifting could be beneficial to enable formal analysis, generalization, and

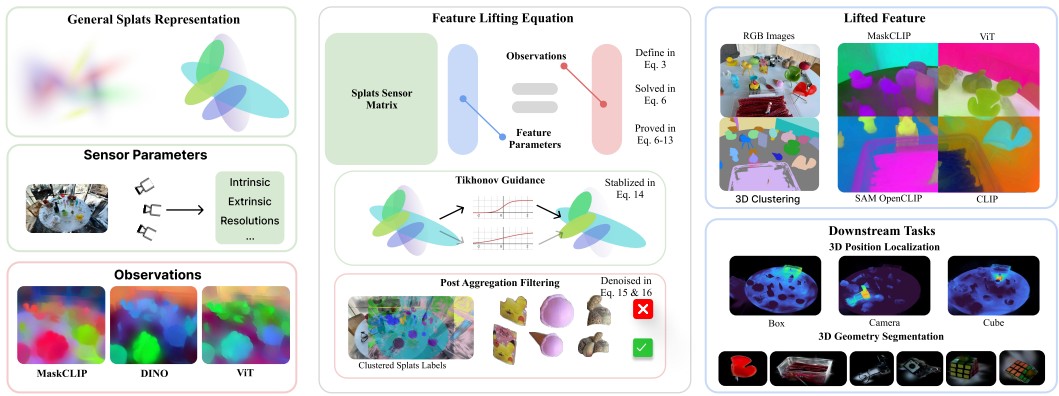

Figure 1: **Overview of our Feature Lifting Framework.** Our pipeline lifts dense 2D feature observations (e.g., MaskCLIP, DINO) onto general 3D splat representations by formulating the task as a sparse linear inverse problem. The **Solver** incorporates Tikhonov Guidance to ensure numerical stability and Post-Lifting Aggregation to filter noisy inputs. The resulting lifted feature parameters enable high-fidelity downstream tasks, such as open-vocabulary 3D segmentation and localization.

optimization. Second, in the absence of a formal definition, existing approaches currently lack theoretical guarantees regarding the quality of the lifted features, leaving some uncertainty about their proximity to a globally optimal solution. Third, all aforementioned methods are exclusively focus on "SAM+CLIP" features and "3DGS" kernels, which may limit generalization across broader settings. Finally, most prior works do not explicitly account for the inherently noisy nature of collected data, where multi-view inconsistencies and observational noise can introduce ambiguity. Our formulation accounts for these challenges through a theoretically grounded linear inverse framework.

Our key contributions are summarized below:

- Propose the formulation of feature lifting as a sparse linear inverse problem, where each per-primitive descriptor is recovered via a global consistent solution to the system $AX = B$
- Prove that, under convex losses and the assumption that our proposed linear system has a unique solution, our solver admits a provable upper bound on the global optimal error
- Introduce two complementary modules to filter noisy input and stabilize the proposed linear system: Tikhonov Guidance and Post-Lifting Feature Refinement
- Provide a generalized implementation for feature lifting to multiple primitive kernels and different dense features
- Achieve state-of-the-art performance on the downstream task of open-vocabulary 3D semantic segmentation.

As illustrated in Figure 1, our framework inputs precomputed splats, sensor parameters, and dense observations. These are mapped to the feature lifting equation $AX = B$, where the Splats Sensor Matrix ($A$) is derived from the geometry and camera parameters, and the observations form the target vector ($B$). To solve for the feature parameters ($X$), we enhance stability by polarizing alpha values to encourage diagonal dominance (Tikhonov Guidance) and employ Post-Lifting Aggregation to discard inconsistent masks. Our formulation is the first to formalize the feature lifting problem as a general linear inverse problem, and can therefore be applied across diverse splat primitives (e.g., 3DGS, 2DGS, Beta Splats) and dense feature modalities (e.g., CLIP, DINO, ViT, ResNet features). Moreover, it establishes a new state-of-the-art on downstream tasks such as open-vocabulary 3D semantic segmentation, outperforming existing training-based, grouping-based, and heuristic-forward baselines in mIoUs.

## 2 RELATED WORK

Prior feature-lifting techniques for 3D tasks can be grouped into three families, each with distinct trade-offs between accuracy and efficiency.

**Joint Training-based Methods**—such as LangSplat Qin et al. (2024), LeGaussian Shi et al. (2024), FMGS Zuo et al. (2025), and FeatureSplats Qiu et al. (2024)—jointly optimize scene geometry and high-dimensional feature embeddings to directly learn splat-wise semantic descriptors. While effective, end-to-end optimization over large descriptor sets and primitives can be computationally intensive, often requiring significant memory and training time. To alleviate resource demands, these approaches employ descriptor compression (e.g., PCA in LangSplat Qin et al. (2024), quantization in LeGaussian Shi et al. (2024), and deep auto-encoders in FeatureSplats Qiu et al. (2024)) or reduce the number of primitives (FMGS Zuo et al. (2025)). These strategies help manage resource usage, though they may introduce trade-offs in terms of geometric detail or feature precision.

**Grouping-based Methods** first extract 2D region or instance masks using models such as SAM Kirillov et al. (2023) and SAM2 Ravi et al. (2024). Following that, each mask is linked to 3D Gaussians via a lightweight training process. Finally, features are aggregated into 3D either by additional optimization or by direct unprojection Wu et al. (2025); Peng et al. (2024b); Gu et al. (2024); Liang et al. (2024); Cen et al. (2025). While these approaches reduce the cost of joint optimization, they still require one to two hours of scene-specific optimization. Furthermore, these methods depend on SAM's per-view masks, while powerful, are designed primarily for instance segmentation and may not be directly suited for noisy, dense, pixel-wise feature lifting.

In particular, LAGA Cen et al. (2025) uses per-view SAM masks Kirillov et al. (2023) and trains an affinity model to cluster 3D Gaussian splats, subsequently associating splat clusters across views. However, this approach assumes that feature variance reflects true semantic differences and is therefore view-dependent, introducing a dynamic K-means clustering step that adds complexity and may increase memory and computational requirements. In contrast, our analysis suggests that such discrepancies often stem from mask inaccuracies—such as one mask isolating only the target while another over-segments and includes adjacent objects—rather than from genuine viewpoint changes. Our convex-regularized inverse problem solver suppresses this noise directly—without requiring additional training, multi-level clustering algorithms, or view-dependent features assigned to one primitive.

**Heuristic Forward Methods**-e.g., Argmax Lifting Chacko et al. (2025), Occam's LGS Cheng et al. (2024), Semantic Gaussian Guo et al. (2024), gradient-guided splitting Joseph et al. (2024), Coseg-Gaussians Dou et al. (2024), and DrSplats Jun-Seong et al. (2025)—are analytic and training-free, offer high efficiency and simplicity. However, they often lack a formal mathematical foundation and sensitive to noisy inputs. Interestingly, three recent heuristic pipelines—CosegGaussians Dou et al. (2024), Occam's LGS Cheng et al. (2024), and DrSplats Jun-Seong et al. (2025)—independently proposed the identical row-sum weighting rule (row-sum preconditioner), underscoring its broad practical effectiveness. However, while CosegGaussians applied the row-sum weighting scheme $x_i = \sum_j w_{i,j} b_j / \sum_j w_{i,j}$ , it did so without an accompanying theoretical interpretation. Occam's LGS later offered a maximum-likelihood interpretation of the same weighted-average rule but provided no error bound. Dr. Splat simplifies the preconditioner further by summing only the top-$k$ contributions per primitive, trading off some theoretical rigor for computational efficiency.

None of the three families of methods explicitly address that feature lifting is fundamentally a **sparse, row-stochastic, linear inverse problem** subject to noise(has misleading masks) and incompleteness—i.e., it is singular and requires careful regularization or preconditioning to ensure stability and obtain a reliable solution approximation. Building on this observation, we model feature lifting as a linear system, which allows us to derive theoretical bounds for any convex loss and demonstrate applicability across a range of embeddings (e.g., DINO, ViT, CNN) and splat's kernels.

## 3 Feature Splat Solver

In this section, we begin by formally defining the feature lifting problem, enabling the transfer of various downstream tasks into a mathematical framework. We then establish optimality bounds for the row-sum preconditioner under the linear inverse problem setting. Finally, we derive two regularization terms to stabilize the solution in the presence of noise, particularly in downstream tasks such as lifting SAM-generated masks embedded with CLIP features.

## 3.1 Problem Definition

### 3.1.1 Splats Rendering

To justify formulating the Feature Lifting problem as a linear inverse problem, we first review the core splats rendering process. Most splats primitives, including planar 2D Gaussian Splats Huang et al. (2024), volumetric 3D Gaussian Splats Yu et al. (2024); Kerbl et al. (2023), and Deformable Beta Splats Liu et al. (2025), employ a fixed, depth-sorted alpha-blending pipeline. Concretely, for each viewing ray $r$, there is a sequence of primitives ordered from front to back, with their respective contribution to the final color denoted as $\omega_p$. The background color is $C_b$. The rendered ray color $C_r$ is given below:

$$\omega_p = \sigma_p \prod_{j=1}^{p-1} (1 - \sigma_j), \sigma_p = \alpha_p \delta_{pr}, \alpha_p = \frac{1}{1 + e^{-\theta}} \tag{1}$$

$$C_r = \sum_p \omega_p c_p + \left(1 - \sum_p \omega_p\right) C_b \tag{2}$$

$\sigma_p$ and $c_p$ denote the opacity and color (or feature) of the $p$-th primitive. For fully opaque primitives (e.g., meshes under standard z-buffering), setting $\forall p, \sigma_p = 1$ reduces Equ.2 to a simple depth test. Here, $\theta$ is the raw opacity term inputted into a sigmoid activation function to produce the final opacity. The primitive difference only affects the kernel function that calculates the $\delta_{pr}$ of ray $r$ on primitive $p$, which is agnostic to the calculation of the final rendered color $C(r)$. There are several notable properties:

**Property 1.** $\omega_p$ *is highly sparse in* $C_r$

**Property 2.** $\sum \omega_p = 1$

**Property 3.** *The rendered color* $C_r$ *is usually very close (i.e. PSNR is more than 24 according to Yu et al. (2024); Kerbl et al. (2023); Liu et al. (2025)) to the given observation* $\hat{C}_r$ *in the training data set.*

The first property arises from the tile-based rendering design of Gaussian SplatsKerbl et al. (2023). Since the tile size is typically set to $16 \times 16$, most splats are excluded from rendering on any given tile.

The second property is the row stochastic property. It is less than one because the standard alpha-blending heuristic of terminating ray integration once the composite opacity approaches one. It is close to one based on the assumption that scene geometry should block any further contribution from background color during training. In practice, we randomize the background color during training, forcing splats to fully occlude the randomly colored background from all viewing directions. Further justification of this property can be found in the appendix.G.

The third observation is noted in a recent splats training method Yu et al. (2024); Huang et al. (2024); Kerbl et al. (2023); Liu et al. (2025). Those properties will be used to justify our formulation.

### 3.1.2 Feature Lifting Equation

Here, we introduce the feature lifting equation.

$$Ax = B, \quad A \in \mathbb{R}^{R \times P}, x \in \mathbb{R}^{P \times F}, B \in \mathbb{R}^{R \times F} \tag{3}$$

$R$ is the number of rays in the observation, $F$ is the dimension of the observation data, and $P$ is the number of primitives. Each row of A represents the observation $w_p$ mentioned in Equ.2 at particular ray $r$. Therefore, we have the following assignment:

$$A_{ij} = \omega_{rp}, \quad x_j = c_p, \quad B_i = \hat{C}_r, \quad (i = r, j = p) \tag{4}$$

Note how $A$ becomes a fixed matrix once the geometry-related attributes are given. Likewise, B becomes fixed once the observation (Features such as CLIP, DINO, or other general features per pixel) has been given. The goal is to solve $x$, or in other words, to determine the feature vector associated with each geometric primitive. We refer to the solution $x$ as the lifted feature representation of primitive $\mathcal{P}$ and observations $\mathcal{O}$.

### 3.1.3 INVERSE PROBLEM

First, we ask whether Equ.3 admits a solution, and, whether the given solution is unique. In typical settings—where $A \in R^{R \times P}$ with $R \gg P$— the system is overdetermined and generally inconsistent, meaning no exact $X$ satisfies equation 3. Instead, we consider the least squares formulation (or more generally, a convex loss minimization problem):

$$X^\star = \arg\min_X \|A\,X - B\|_F^2, \tag{5}$$

where the minimizer exists for any $B$ and lies in the column space of $A$. In our context, Property 3 guarantees that real-world observations $B$, such as RGB color, approximately lie within $\text{range}(A)$. As a result, the least-squares residual is small, and a near-exact lift exists. Intuitively, any observation derived from the same RGB signal should have a near-exact lift. From a semantic viewpoint, we also require a one-to-one mapping. Each geometric primitive should admit a single descriptor, mirroring an ideal embedding (e.g. a perfect 3D CLIP) that assigns one feature per object. Thus, uniqueness in equation 5 is not just for mathematical convenience, but for the purpose of aligning with downstream task requirements. Any deviation from uniqueness signals is either due to noisy inputs or suboptimal solvers—precisely the singular, and misleading-mask problem we address through our convex regularization and preconditioning strategies.

When the observed signal varies smoothly with the camera parameters (intrinsics and extrinsics), the inverse problem satisfies the afore-mentioned three-pronged criteria: a solution *exists*, is *unique*, and depends *continuously* on the data. However, when the signal is discontinuous or corrupted by noise— for example, if one view's segmentation mask captures only the noodles of a ramen bowl, while the next view's mask includes both the bowl and noodles—the resulting CLIP embeddings jump abruptly. In such cases, there is no exact solution. The optimal "solution" will be the least square "solution". our proposed solver then served as a bounded approximation to the least square solution. Our proposed regularizer will filter out the misleading signals.

### 3.2 SOLVER

In this session, we will introduce our solver under the well-posedness condition and give an optimal upper bound of our solver. In our setting, the least-squares problem in Eq. equation 5 is convex, so stochastic methods such as SGD provably converge to the global minimizer Esser et al. (2010); Saad (2003). However, cold-start training can be prohibitively slow. Prior works (e.g. Chacko et al. (2025); Dou et al. (2024); Jun-Seong et al. (2025); Cheng et al. (2024); Joseph et al. (2024)) have explored efficient one-shot approximations that show practical value but do not offer formal guarantees. To address this, we introduce the row-sum preconditioner as a, closed-form solver. We then develop an analytical framework that (i) quantifies its $(1 + \beta)$-approximation error under both L2 and general convex losses, and (ii) interprets the behavior of existing heuristics as special cases.

$$D^{\frac{1}{2}}(A^T A) = \sqrt{D(A^T A)} \quad x = D^{-\frac{1}{2}}(A^T A)e \times D^{\frac{1}{2}}(A^T A)B \quad x_j = \frac{\sum_i A_{ij} B_i}{\sum_i A_{ij}} \tag{6}$$

The proposed initial solution is given in Equ.6, where $D^{\frac{1}{2}}$ is the diagonal operator containing square roots of the row sums, and $e$ is all-ones vector. The expression on the right-hand side is the element-wise formulation. We now proceed to analyze the optimality of the proposed solution.

$$\mathcal{L}(x) = \sum_{i=1}^R \left\| \sum_{j=1}^P A_{ij} x_j - B_i \right\| \quad \mathcal{J}(x) = \sum_i \sum_j A_{ij} \|x_j - B_i\| \tag{7}$$

$$\sum_j A_{ij} = 1 \quad \Rightarrow \quad \mathcal{L}(x) \overset{\text{Jensen}}{\leq} \mathcal{J}(x) \tag{8}$$

Here, we start with the original loss function $\mathcal{L}(x)$, and calculate a surrogate loss function $\mathcal{J}(x)$. The norm $\|\cdot\|$ represents any convex loss function such as L1, L2, or a Huber loss. From Property.2, we apply the Jensen's inequality. This means the surrogate loss function $\mathcal{J}$ is larger than $\mathcal{L}$ and serves as an upper bound for the true loss. For the sake of argument, we consider a special case where $\|\cdot\|$ is the L2 loss. If $\mathcal{J}$ attains a minimum, it occurs where the gradient is zero, as shown in Equ.9.

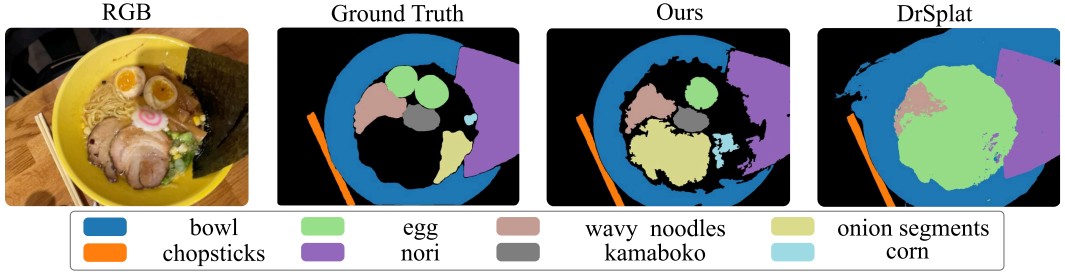

Figure 2: **Qualitative comparison on the LeRF-OVS Ramen scene.** We compare our method against DrSplats and the Ground Truth. As shown in the legend, distinct colors represent different semantic classes (e.g., egg, chopsticks). Our method performs better compared to recent SOTA Jun-Seong et al. (2025). More qualitative result could be found in Fig. 3, Fig. 4, and Fig. 5

Therefore, we obtain an optimal solution on the surrogate loss function $\mathcal{J}$.

$$\frac{\partial \mathcal{J}}{\partial x_j} = \sum_i A_{ij} \left(x_j - B_i\right), \quad \frac{\partial \mathcal{J}}{\partial x_j} = 0 \quad \Rightarrow \quad x'_j = \frac{\sum_i A_{ij} B_i}{\sum_i A_{ij}} \tag{9}$$

We now define $\beta$, which intuitively measures the dispersion of the lifted feature along a viewing ray at the global optimal lift. Suppose the optimal lift yields a solution $\hat{x}$. We define $\beta$ at Equ.11.

$$\Delta_{ij} = \|\hat{x}_j - B_i\|, \quad \mu_i = \sum_j A_{ij} \Delta_{ij} \tag{10}$$

$$\sigma_i^2 = \sum_j A_{ij} \left(\Delta_{ij}^2 - \mu_i^2\right), \beta_i = \frac{\sigma_i^2}{\mu_i^2}, \beta = \max_i(\beta_i) \tag{11}$$

**Property 4** (Diagonal Dominance Reduces $\beta$). *If each row of $A$ becomes increasingly diagonally dominant—i.e., one entry in each row satisfies $A^T A_{jj} \gg A^T A_{jk}$ for all $k \neq j$—then the dispersion $\beta$ is smaller. In other words, stronger diagonal dominance results in a smaller beta.*

Notice that usually $x' \neq \hat{x}$. Therefore, $x'$ is not optimal in $\mathcal{J}$. By Equ.8, we have the following equation under the L2 loss:

$$\mathcal{L}(x') \leq \mathcal{J}(x') \leq \mathcal{J}(\hat{x}) \tag{12}$$

$$\mathcal{J}(\hat{x}) = \sum_i \sum_j A_{ij} \Delta_{ij}^2 \sum (1 + \beta_i) \mu_i^2 \leq (1 + \beta)\mathcal{L}(\hat{x}) \Rightarrow \mathcal{L}(x') \leq (1 + \beta)\mathcal{L}(\hat{x}) \tag{13}$$

While our $1 + \beta$-approximation is proven under the L2 loss for any convex loss $l(\cdot)$ that is Lipschitz-smooth or strongly convex, the proposed solution is bounded above by a function of $\beta$. This generalizes the stability and approximation behavior of our method beyond the Euclidean norm. To be more specific, the Jensen equation always hold if we are using any convex loss function. While it is not strictly bounded by the $1 + \beta$, one can always find a finite transform of $1 + \beta$ that is bounded according to proposed loss function.

From the analysis, we can conclude that the methods inDou et al. (2024); Cheng et al. (2024) are also bounded by the $1 + \beta$ term. The method Chacko et al. (2025) is bounded when a $l_\infty$ norm function is applied. The approach in Joseph et al. (2024) is bounded only when the input observations are unit vectors. Jun-Seong et al. (2025) does not provide a theoretical bound, as it selects the top-K splats based solely on depth ordering. Although our solver can lift any 2D feature into high-dimensional primitives, few benchmarks exist to evaluate such capability. We evaluate our approach on the LeRF-OVS Qin et al. (2024) and 3D-OVS Liu et al. (2023) datasets, using the segmentation masks annotated from Qin et al. (2024); Kerr et al. (2023), and report mean Intersection-over-Union (mIoU) according to the protocols of Cen et al. (2025); Jun-Seong et al. (2025). As Dr. Splat's official code was not available at the time of writing, we follow the evaluation protocol provided by LAGA for consistency. CosegGS and Occam's LGS Dou et al. (2024); Cheng et al. (2024), both based on the same naïve solver (Eq. 6), are represented here by a single baseline. As shown in Tab.1, we group the methods into four categories: 2D- and NeRF-based, training-based, grouping-based, and heuristic forward-lifting. First, second, and third place in the benchmark are highlighted in light red, light pink, and light yellow, respectively. Results for the 3D-OVSLiu et al. (2023) dataset is provided in the appendix.F

### 3.3 REGULARIZER

We have shown that our solution is $\beta$-bounded. However, because this approximation does not rely on any continuity assumptions, it can become quite coarse in the presence of strong noise. In particular, if the linear operator $A$ is rank-deficient (or nearly so) and thus fails to satisfy continuity requirements, the resulting solution will be noisy. To address this, we introduce two regularization strategies: one to regularize the operator $A$ itself, and another to filter the observations $B$. By removing noisy components and reinforcing diagonal dominance, our solver produces more stable feature-lifting results in downstream tasks.

#### 3.3.1 TIKHONOV GUIDANCE

According to property.4 and inspired by A. & V. (1977); A. (1963), we observe that emphasizing diagonal dominance helps mitigate noise from linear system A. Therefore, we propose a carefully designed regularizer applied during solving without sacrificing the RGB rendered result. Specifically, we first convert Equ.6 into a fully diagonal row-sum version. . Briefly speaking, stabilizing the linear system $A^T A$ involves avoiding small eigenvalues. Mathematically, by adding a diagonal matrix ,which is strictly non-singular to the original $A^T A$ as shown in Equ.14, we obtain matrix $\tilde{A}$ with larger eigenvalues. Intuitively, one extreme way to decrease the value of $\beta$ would be to make splats either transparent, or fully opaque. In this extreme case, each row in $\tilde{A}$ would contain only a single non-zero entry with a value of one, yielding a globally optimal solution. We employed such regularization by carefully adjusting the opacity activation term during feature lifting. Compare to a linear adjustment like original Tikhonov Regularizer, our method utilizes a non-linear soft guidance without undermine the visual quality. We introduce more implementation details about the Tikhonov Guidance in the appendix.B.

$$\min \left( ||A\tilde{x} - b||^2 + ||\lambda I||^2 \right) \tag{14}$$

#### 3.3.2 POST AGGREGATION FILTERING

Inspired by LAGA Cen et al. (2025), we cluster the lifted features to identify and remove noise from SAM-generated masks, as shown as Equ.15, where -1 denotes unclassified splats, and $K + 1$ is the number of clusters. Unlike LAGA, which learns separate affinity features, we simply reuse the Tikhonov-Guided solution $\tilde{x}$ from Eq. 14 as our clustering feature. This immediately assigns each splat to a cluster. We then encode each Gaussian's cluster ID as a one-hot signal and render this signal back to 2D. By applying an argmax operation, we recover a 2D mask for each cluster, establishing a pixel-level correspondence between the clusters and the original SAM masks as shown in Equ.15 and Equ.15 (Note: Any orthogonal encoding-decoding System could be used here).

The remaining masks—denoted as $B'$—are significantly more self-consistent and better aligned with the underlying decomposition of splat primitives. The visualization of the alignment is displayed in the technical appendix.C. This produces a well–posed data set that further boosts downstream semantic-segmentation and object-querying performance to state-of-the-art levels. In contrast to LAGA's complex view-dependent clustering, our simpler Post Aggregation Filtering approach reveals that most apparent "view-dependent" variations stem from mask noise rather than useful multi-view information. Noisy masks can be found in Fig.1 as well as Fig.7. We further provide many mis-leading masks' visualization and trust worthy masks in the appendix.C.

$$\gamma = \text{Agg}(x) \in \{-1, \dots, K\}^P, \quad \Gamma = \text{onehot}(\gamma) \in \{0, 1\}^{P \times (K+1)}, \tag{15a}$$

$$\kappa = \arg\max(A\Gamma) - 1 \in \{-1, \dots, K\}^R, \tag{15b}$$

The function $M$ identifies the binary mask corresponding to ray $r_j$ based on the original observation in $B$, and similarly finds the mask for the projected label $\kappa$ associated with the same ray $r_j$. More explicitly, the function $M$ gathers all pixels within the same image that share the same label and constructs a binary mask. We then calculate the Intersection over Union (IoU) between each SAM mask and its corresponding cluster mask. Masks with an overlap below a predefined threshold are discarded, as detailed in Equation 16.

$$M(j, B) = m_j \in \mathbb{R}^{H \times W}, M(j, \kappa) = m'_j \in \mathbb{R}^{H \times W}, \quad B'_j = \begin{cases} B_j, & \text{IoU}(m_j, m'_j) \geq \tau, \\ \emptyset, & \text{otherwise.} \end{cases} \tag{16}$$

Table 1: **LeRF OVS (left) vs. Multi-Feature & Ablation (right).**

(a)**IoU comparison on LeRF OVS**. The symbol † denotes results reported directly from the original papers, while ‡ indicates results evaluated using their official implementations. Our mIoU evaluation follows the LAGACen et al. (2025) implementation. For a fair comparison, we use the same 3D Gaussian model as the lifting primitive, trained using the official repository provided by LAGACen et al. (2025). The same trained model is used for LAGA, DrSplat, Occam'LGS, and our method in all ‡ evaluations. F denotes Figurines, T is Teatime, R is Ramen, W is Waldo kitchen. We use SAM+OpenCLIP for encoding for fair comparison

| Method/Scene | F | T. | R. | W. | Mean |
|---|---|---|---|---|---|
| LSeg† Li et al. (2022) | 7.6 | 21.7 | 7.0 | 29.9 | 16.6 |
| LeRF† Kerr et al. (2023) | 38.6 | 45.0 | 28.2 | 37.9 | 37.4 |
| N2F2† Bhalgat et al. (2024) | 47.0 | 69.2 | 56.6 | 47.9 | 54.4 |
| LangSplat† Qin et al. (2024) | 25.9 | 35.6 | 29.3 | 33.5 | 31.1 |
| LeGaussian† Shi et al. (2024) | 31.2 | 34.5 | 17.6 | 17.3 | 25.2 |
| SuperGseg† Liang et al. (2024) | 43.7 | 55.3 | 18.1 | 26.7 | 35.9 |
| VLGS† Peng et al. (2024a) | 58.1 | 73.5 | 61.4 | 54.8 | 62.0 |
| SAGA† Cen et al. (2023) | 36.2 | 19.3 | 53.1 | 14.4 | 30.7 |
| OpenGaussian† Wu et al. (2025) | 61.1 | 59.1 | 29.2 | 31.9 | 45.3 |
| GS Grouping† Ye et al. (2023) | 60.9 | 40.0 | 45.5 | 38.7 | 46.3 |
| LAGA‡ Cen et al. (2025) | 56.1 | 68.9 | 57.4 | 64.6 | 61.7 |
| LAGA† Cen et al. (2025) | 64.1 | 70.9 | 55.6 | 65.6 | 64.0 |
| DrSplat‡ Jun-Seong et al. (2025) | 47.5 | 66.2 | 36.7 | 47.5 | 49.5 |
| DrSplat† Jun-Seong et al. (2025) | 53.4 | 57.2 | 24.7 | 39.1 | 43.6 |
| OccamLGS‡ Cheng et al. (2024) | 60.1 | 68.3 | 55.3 | 47.7 | 57.8 |
| OccamLGS† Cheng et al. (2024) | 58.6 | 70.2 | 51.0 | 65.3 | 61.3 |
| Ours | 67.6 | 68.5 | 62.3 | 62.1 | 65.1 |

(b)**Multi Feature Comparison on Cosine-Similarity**. We evaluate the feature-agnostic capability without Tikhonov guidance, post-aggregation filtering, or automatic threshold selection. We apply our method to dense features generated by FeatUp Fu et al. (2024), including features from DINO, DINOv2, CLIP, MaskCLIP, ViT, and ResNet. For a fair comparison, we use geometry provided by Gsplat to obtain the geometry for all scenes. Notice that SAM + OpenCLIP is follow exactly the LangSplatQin et al. (2024) implementation.

| Features/Scene | F. | T | R | W. | Mean |
|---|---|---|---|---|---|
| SAM+OpenCLIP Kirillov et al. (2023) | 89.3 | 90.7 | 90.6 | 91.0 | 90.4 |
| MaskCLIP Dong et al. (2023) | 92.4 | 94.0 | 94.5 | 94.7 | 93.9 |
| CLIP Radford et al. (2021) | 91.9 | 93.7 | 94.6 | 94.9 | 93.8 |
| DINO Caron et al. (2021) | 78.7 | 80.2 | 83.0 | 82.0 | 81.0 |
| DINOv2 Oquab et al. (2023) | 83.5 | 85.6 | 89.6 | 89.2 | 87.0 |
| ViT Dosovitskiy et al. (2020) | 84.6 | 83.7 | 86.4 | 88.0 | 85.7 |
| ResNet He et al. (2016) | 95.8 | 94.8 | 96.2 | 97.0 | 96.0 |

(c)**Ablation Study**. We evaluate the impact of each module on the LeRF data in a step-by-step manner. $T_i$ denotes Tikhonov Guidance, P denotes Post Aggregation Filtering, and A represents Automatic threshold selection. The baseline corresponds to the row-sum formulation in Equ.18. To further understand which components is effective We provide a full ablation study shown in the appendix, Tab.4.

| Method | F. | T | R | W. | Mean |
|---|---|---|---|---|---|
| Ours w/o ($T_i$PA) | 60.1 | 68.3 | 55.3 | 47.7 | 57.8 |
| Ours w/o (PA) | 61.7 | 67.8 | 53.8 | 49.6 | 58.2 |
| Ours w/o ($T_i$A) | 65.5 | 72.0 | 58.6 | 50.4 | 61.6 |
| Ours w/o (A) | 64.8 | 71.6 | 61.7 | 54.7 | 63.2 |
| Ours (full) | 67.6 | 62.3 | 68.5 | 62.1 | 65.1 |

Table 2: **Multi-Kernel and Runtime Comparison.** Left: Multi-Kernel across different splats. Right: end-to-end runtime.

(a)**Multi Kernel Comparison on mIoU**. For this comparison, we use our method without Tikhonov-Guidance, Post Aggregation Filtering, or auto threshold selection to illustrate its kernel agnostic capability. We evaluate performance using DBS, 2DGS, as well as 3DGS, # represents the inriaKerbl et al. (2023) implementation tuned hyper parameters exactly according to LAGACen et al. (2025) released code, * represents GsplatTancik et al. (2023) implementation with no additional hyper parameters changed except set the background color to random.

| Method | F. | T | R | W. | Mean |
|---|---|---|---|---|---|
| DBS Liu et al. (2025) | 49.5 | 50.8 | 51.2 | 61.3 | 53.2 |
| 3DGS* Tancik et al. (2023) | 55.3 | 63.5 | 47.8 | 49.8 | 54.1 |
| 3DGS# Kerbl et al. (2023) | 60.1 | 68.3 | 55.3 | 47.7 | 57.8 |
| 2DGS* Huang et al. (2024) | 62.0 | 66.3 | 56.0 | 51.1 | 58.9 |

(b)**Runtime Comparison**. We compare the runtime of our lifting method with LAGA and DrSplats, along with the ability to preserve the full feature dimensionality. Here $T_i$ denotes Tikhonov Guidance, P denotes Post Aggregation Filtering. All timings are measured on a single RTX 4090 GPU using CUDA Toolkit 12.8 for consistency. Notice that the runtime of lifting-based methods scales linearly with the number of views, whereas training-based and grouping-based methods are determined by predefined training steps. For DrSplats, we report results using Top-40 follow their implementation.

| Method | F. | T | R | W. | Mean |
|---|---|---|---|---|---|
| Ours w/o ($T_i$P) | 00:03:29 | 00:02:06 | 00:01:28 | 00:02:13 | 00:02:12 |
| Ours w/o (P) | 00:03:14 | 00:02:05 | 00:01:12 | 00:01:47 | 00:02:05 |
| Ours full | 00:05:22 | 00:02:37 | 00:01:59 | 00:03:02 | 00:03:15 |
| DrSplat | 00:02:55 | 00:01:19 | 00:01:03 | 00:01:33 | 00:01:43 |
| LAGA | 01:43:33 | 01:23:22 | 01:20:48 | 01:30:16 | 01:29:30 |

### 3.3.3 Auto Threshold Selection

We observe that threshold selection can significantly alter the final result. Rather than manually choosing thresholds per object (as in Qin et al. (2024)), we leverage the high contrast of the raw attention map for automatic selection. As shown in Fig. 15, even though the ideal hard threshold for correctly isolating chopsticks shifts from 0.225 to 0.30 in the raw attention values, the first valley in the attention histogram remains clearly identifiable. We therefore derive our threshold directly from the histogram's local extrema. We first locate its largest peak and then take the adjacent, smaller valley as the threshold. Further details are provided in the Appendix.I

## 4 EXPERIMENT

We demonstrate our kernel-agnostic capability with a multi-kernel comparison Tab.2. In this study, the 2DGS implementation outperforms all other representations due to its specially designed kernel that encourages high opacity values, yielding a naturally lower $\beta$ as a result.

For downstream segmentation tasks, qualitative results in Fig.2, Fig.5, Fig. 3 and Fig. 4 demonstrate that our method generates significantly clearer and more localized attention maps for object queries.

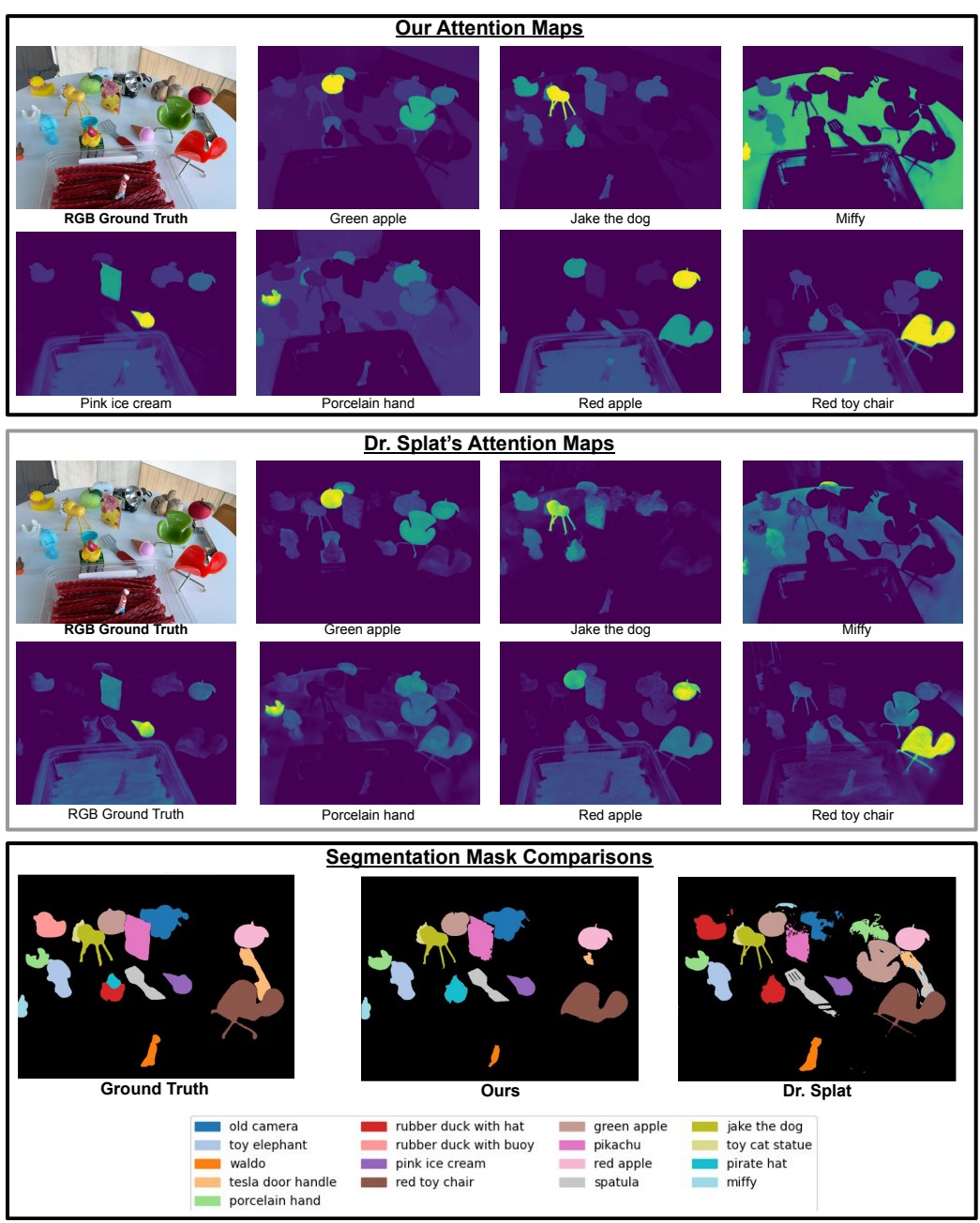

Figure 3: **Qualitative Comparison of Attention Maps and Segmentation Masks.** We present a qualitative comparison between our proposed method (top row) and the baseline, Dr. Splat (middle row). The heatmaps visualize the model's attention response, or affinity feature projection, to various text prompts (e.g., "Green apple," "Jake the dog," "Miffy"). The bottom row compares the final semantic segmentation masks for the entire scene against the ground truth.

In contrast, while the baseline (Dr. Splat) identifies the general location of objects, its activations are more diffuse and less uniform. This lack of precision leads to lower performance in downstream segmentation. As observed, our focused attention maps translate into segmentation masks that closely match the ground truth, exhibiting fewer false positives and less noise than the baseline. These qualitative observations align well with the quantitative mIoU results reported in Table 1.

To compare feature lifting quality more broadly, we project a variety of descriptorsFu et al. (2024)—MaskCLIP, CLIP, ViT, DINO, and ResNet—through our solver (prior to any post-aggregation filtering) and visualize the results using cosine similarity and PCA. Tab.9, Fig.13 and Fig.14 show consistent improvement on current SOTA lifting methodsGuo et al. (2024); Chacko et al. (2025). (see the appendix.H for details). We avoid PSNR for evaluation since MSE-based metrics scale with the number of feature channels. For example, DINO features (384 channels) yield consistently lower PSNR than that of CLIP features (512 channels) despite having comparable perceptual quality. Instead, we demonstrate both quantitatively via cosine similarity as mentioned in Tab.1, and qualitatively via PCA visualization that our lifted projections consistently outperform the raw descriptors, making direct PSNR comparisons potentially misleading. While we conduct experiments with different types of primitives—including 2DGS, 3DGS, and DBS, for comparing feature descriptors we use only 3DGs as the standard primitive to ensure fairness.

To compare runtimes, we report times for lifting, and post-processing using standard SAM+OpenCLIP as the encoder as mentioned in Tab.2. Additionally, we perform further experiments using MaskCLIP as an encoder, which are included in the appendix.E. The motivation behind this choice is that standard SAM+OpenCLIP typically takes approximately one hour to pre-process the dataset, whereas MaskCLIP completes the same pre-processing task within minutes. Additional mIoU results for MaskCLIP are also provided in the appendix.E. The splats training time per scene is roughly 20 minutes for the Gsplats Tancik et al. (2023) training pipeline and approximately 40 minutes for the Inria-based pipeline. Regarding memory consumption, our implementation can lift features with over 512 channels without encountering CUDA out-of-memory (OOM) errors. In contrast, in our testing environment, DrSplat's implementation could only handle up to 32 channels before hitting memory limitations. Lifting-based methods exhibit linear complexity with respect to the number of feature channels. , view images, and witness splats per ray.

Finally, our ablation study isolates the impact of each component—naïve preconditioner, full Tikhonov (as shown in Table 4)—and post-aggregation filtering. We also examine implementation details, including feature scaling, multi-level SAM mask integration, and segmentation-threshold selection. All qualitative results and additional engineering notes are provided in the appendix.E

## 5 DISCUSSION

Our method provides a framework to distinguish meaningful 3D clusters from noise, offering a foundation for building more robust feature lifting pipelines in 3D scene understanding. In practice, the noises and inconsistency in sensing process also poses a challenge in feature lifting process. Our method addresses such issues via regularization and could inform the development of sensing mechanisms that yield more consistent observations.

From a theoretical standpoint, averaging features across views helps suppress view-dependent components while preserving view-independent structures. Such view dependence information can then be quantified by subtracting the lifted (3D) features from the original 2D features to obtain any structured residual patterns correspond to view-specific information.

Finally, while our method focuses on lifting dense features, it will also be feasible to lift sparse descriptors (e.g., SIFT) by embedding them into a dense representation using auxiliary signals. That being said, visualizing sparse 3D splats remains an open challenge.

## 6 CONCLUSION

In this work, we formulate feature lifting as a sparse linear inverse problem and derive a general approximation to its core equation. We prove that our solution achieves a globally bounded error. To further refine the reconstructed features, we introduce two complementary modules: (1) Tikhonov Guidance and (2) Post-Lifting Aggregation. Our implementation completes the lifting process in under 10 minutes.

The method is designed to be broadly applicable to any dense feature representation and splat-based kernel. Experiments on multiple 3D semantic segmentation benchmarks demonstrate state-of-the-art performance. We also quantitatively evaluate the quality of lifted features using cosine similarity metrics to demonstrate the method's effectiveness across different feature descriptors. Together, the method provides a scalable and theoretically framework for enrichment of 3D representations with 2D dense features to support scene understandings and other downstream tasks.

## 7 ACKNOWLEDGMENT

The authors would like to thank our primary sponsors of this research: Mr. Clayton Burford of the Battlespace Content Creation (BCC) team at Simulation and Training Technology Center (STTC). This work is supported by University Affiliated Research Center (UARC) award W911NF-14-D-0005. Statements and opinions expressed and content included do not necessarily reflect the position or the policy of the Government, and no official endorsement should be inferred.

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

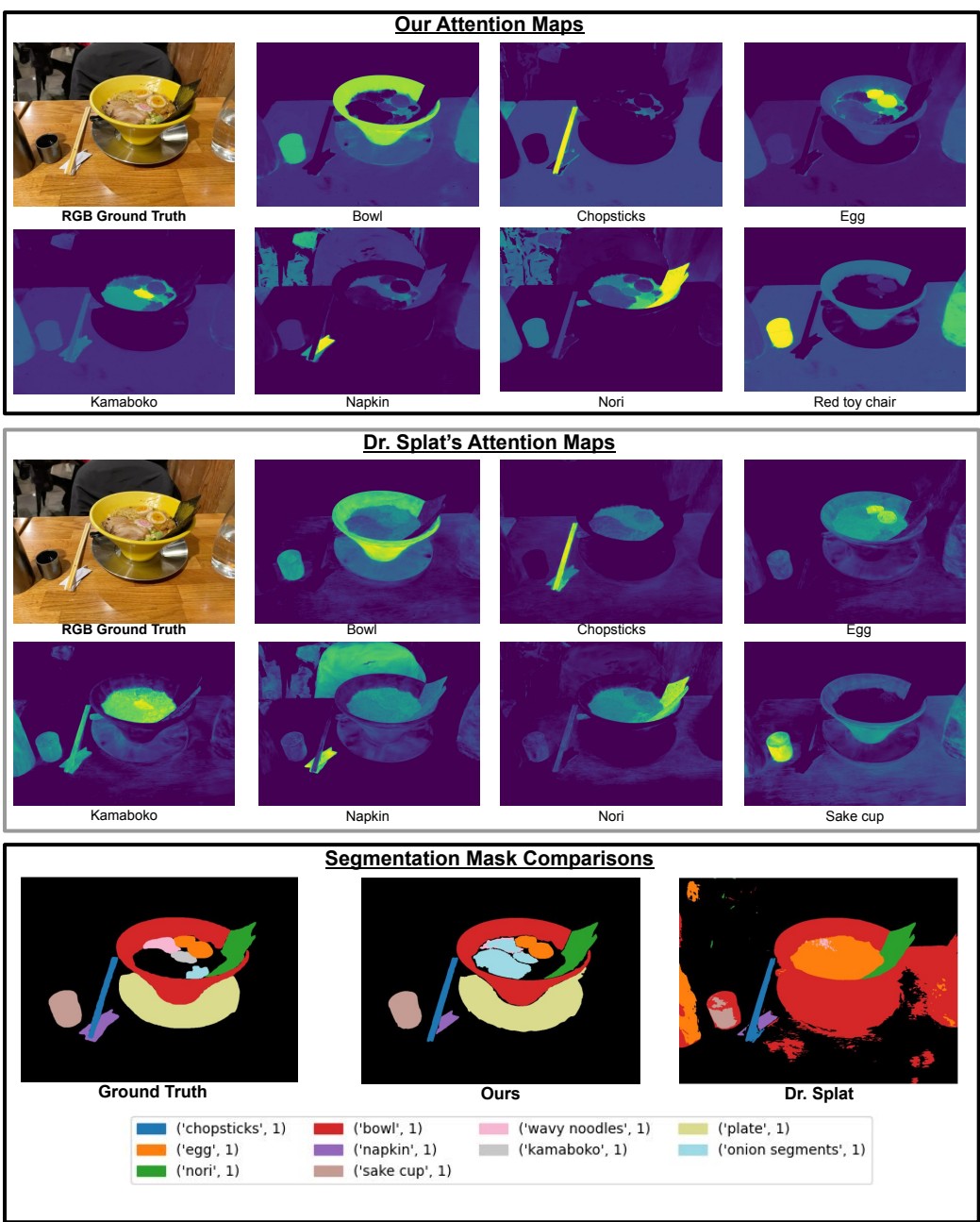

Figure 4: **Attention Map Comparison:** As demonstrated in the figures, our implementation produces clearer attention maps and segmentation masks compared to Dr. Splat.

## A    MULTIMEDIA SUPPLEMENTARY

In the supplementary we submitted, we have additional embedding website. The **website** is local, so one needs to open it by uncompressing the website package and click the index.html. We have additional **code** zip file with detail README guidance. To preview the overall result, one can ckeck our **video**.

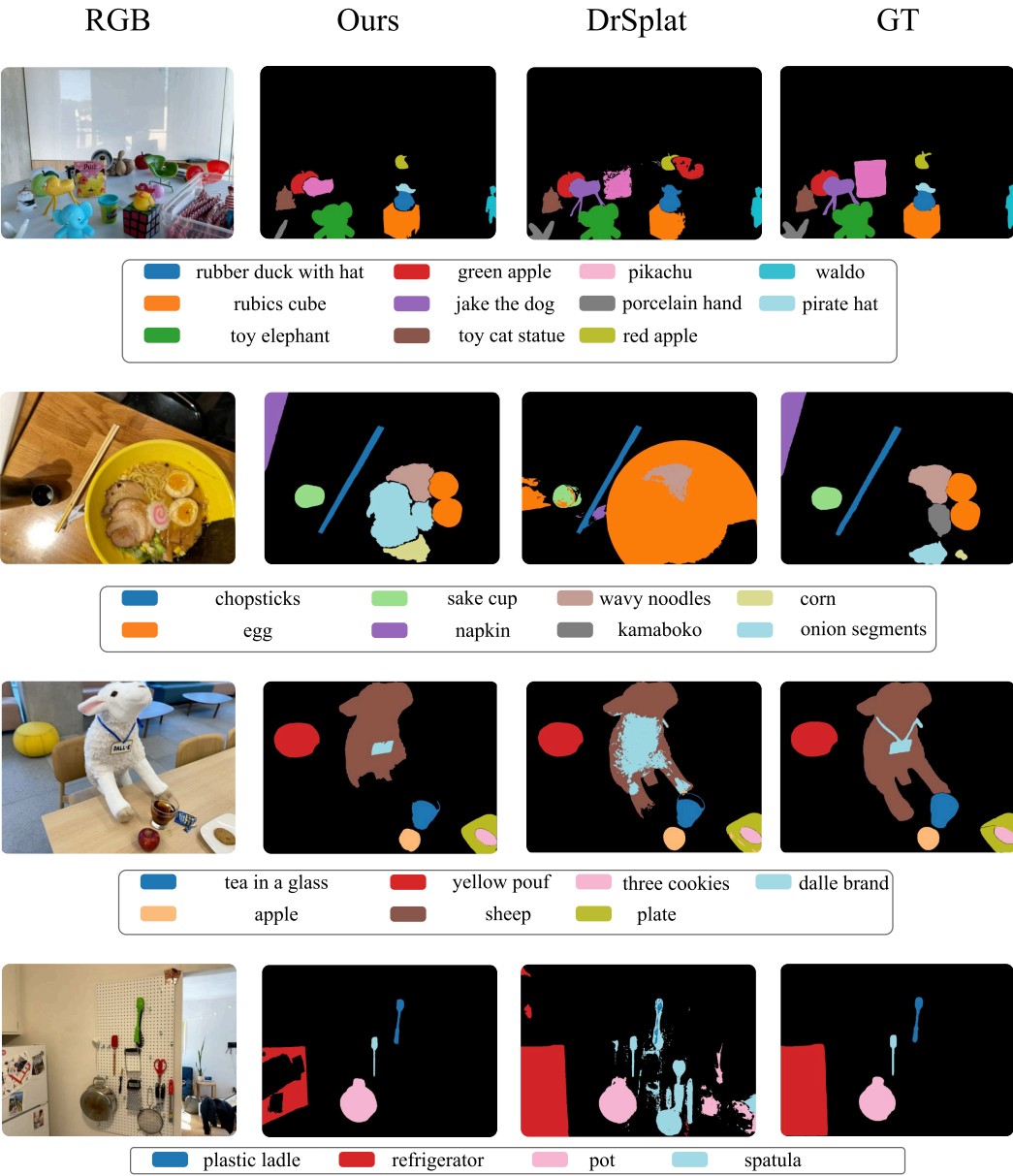

Figure 5: **Additional Qualitative Segmentation Results.** Scenes from top to bottom: Figurines, Ramen, Teatime, and Waldo's Kitchen (LeRF-OVS dataset). Our method consistently produces sharper, less noisy segmentation masks compared to DrSplat and aligns closely with the Ground Truth.

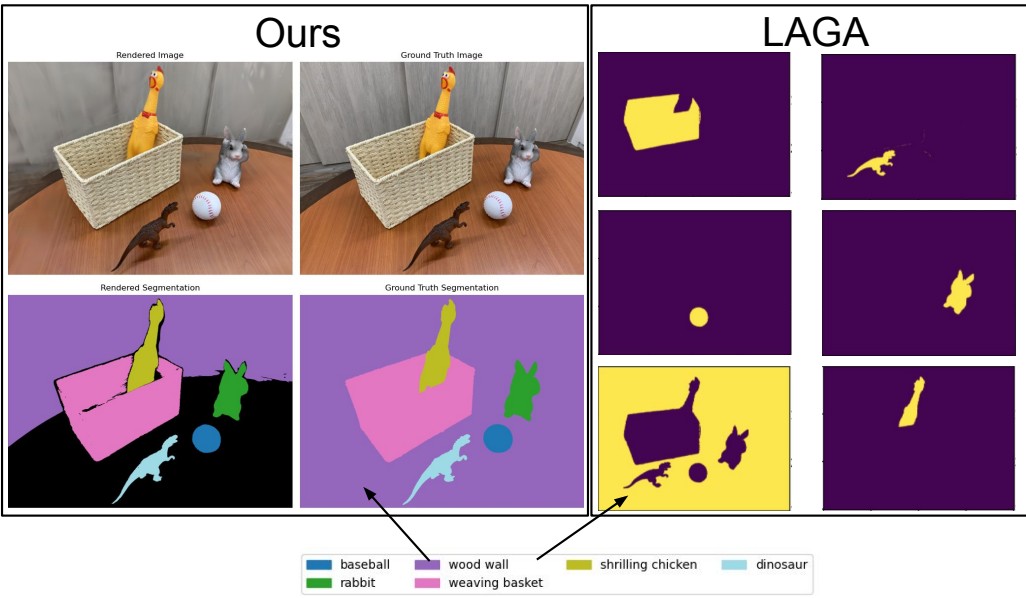

Figure 6: The left column shows the results from our method, while the right column (yellow-purple mask) shows the results generated by LAGA for the 3DOVS 'room' scene. Although the 3DOVS ground truth considers the wooden table as part of the wooden wall, our model is able to clearly distinguish between the two. While LAGA's result appears to achieve a higher mIoU on this particular scene, it actually generates misleading masks.

## B TIKHONOV GUIDANCE

First, let us discuss why Tikhonov regularization is **theoretically** important for stability, and why our system can become unstable. Instability arises when the matrix $\mathbf{A}$ contains nearly identical rows. In our setting, adjacent pixels often undergo almost the same rendering and alpha-blending processes, leading to duplicated weights. Redundant rows with similar observation values typically pose no problem (as in the RGB domain), but if identical rows correspond to different observations, the system becomes singular and no exact solution exists. Even nearly identical rows introduce large errors in $\mathbf{B}$ due to near-singularity in $\mathbf{A}$.

Mathematically, a common remedy is to strengthen the diagonal. Specifically, we first convert Equation 6 into a fully diagonal, row-sum preconditioner by squaring the original weights. We then adjust the opacity activation term $\lambda$ by compressing the sigmoid function to polarize each splat's opacity. Because lifting is performed without modifying any geometry-training parameters, this approach does not compromise image-based evaluation metrics. We choose $\lambda = 1.2$ for both the lifting equation and the attention-map projection, as shown by the results in Table 6.

There are a few notes about the polarization. To be specific, after we get the original training figure, the first thing we need to do is to convert it to a Regularized version, by injecting our $\lambda$ in this stage, we get a polarized alpha value. And after lifting, when we do the 3D segmentation, or localization, we can directly use the feature generated. But the challenge is, when we project the attention queried back to 2D to do the segmentation, what alpha should we use. Experiments shows that, using the same $\lambda$ is the best for the 2D attention map generation.

$$\tilde{A}_{ij} = \tilde{\omega}_{ij}, \quad \tilde{\alpha}_p = \frac{1}{1 + e^{-\lambda\theta}} \tag{17}$$

$$x = D^{-1}(\tilde{A}^T\tilde{A})e \times D(\tilde{A}^T\tilde{A})B \quad x_j = \frac{\sum_i \tilde{A}_{ij}^2 B_i}{\sum_i \tilde{A}_{ij}^2} \tag{18}$$

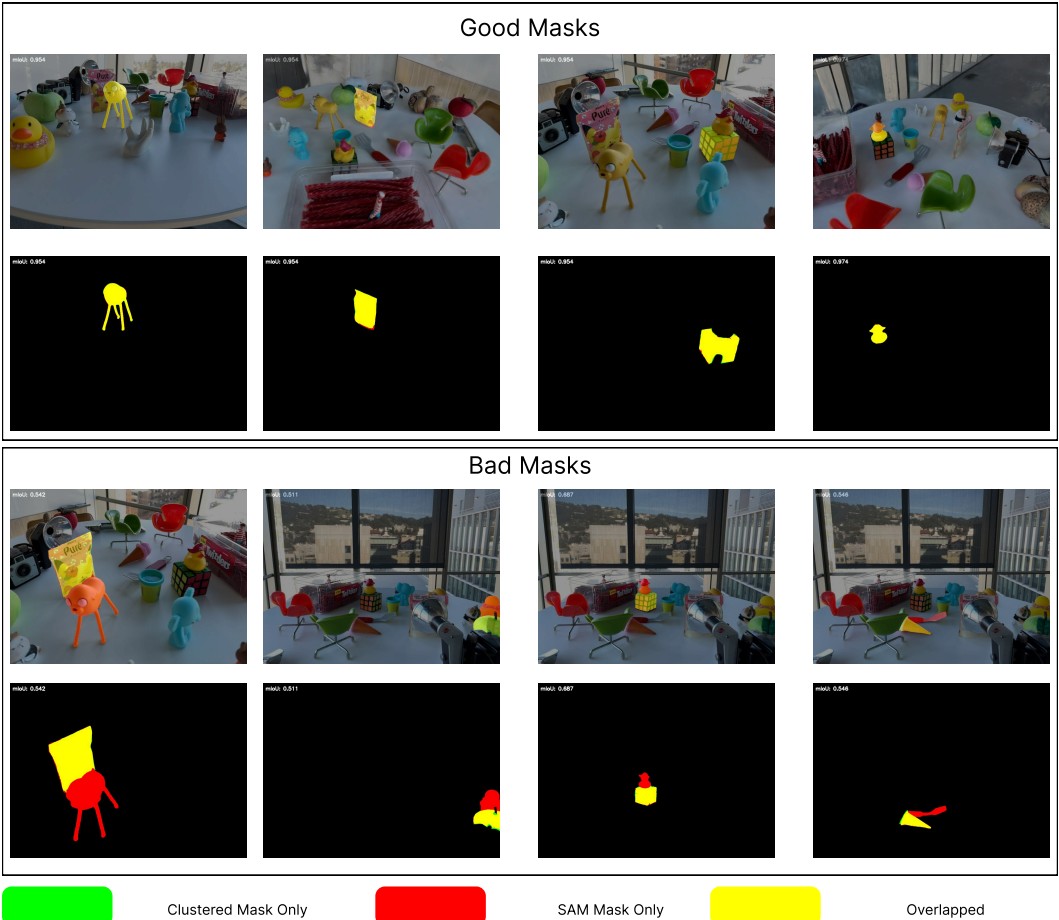

Figure 7: Here we display the figure for post lifting aggregation examples. The first row is displaying the right mask generated in the SAM, while the second row is the mis-leading masks generated in the SAM model. We compare our clustered masks displayed in green with the SAM generated mask which displayed in red. And we will filter out the low yellow region (overlapping region) masks through mIoU thresholding. This observation shows that view-dependent information might due to the mis-leading masks generated by SAM.

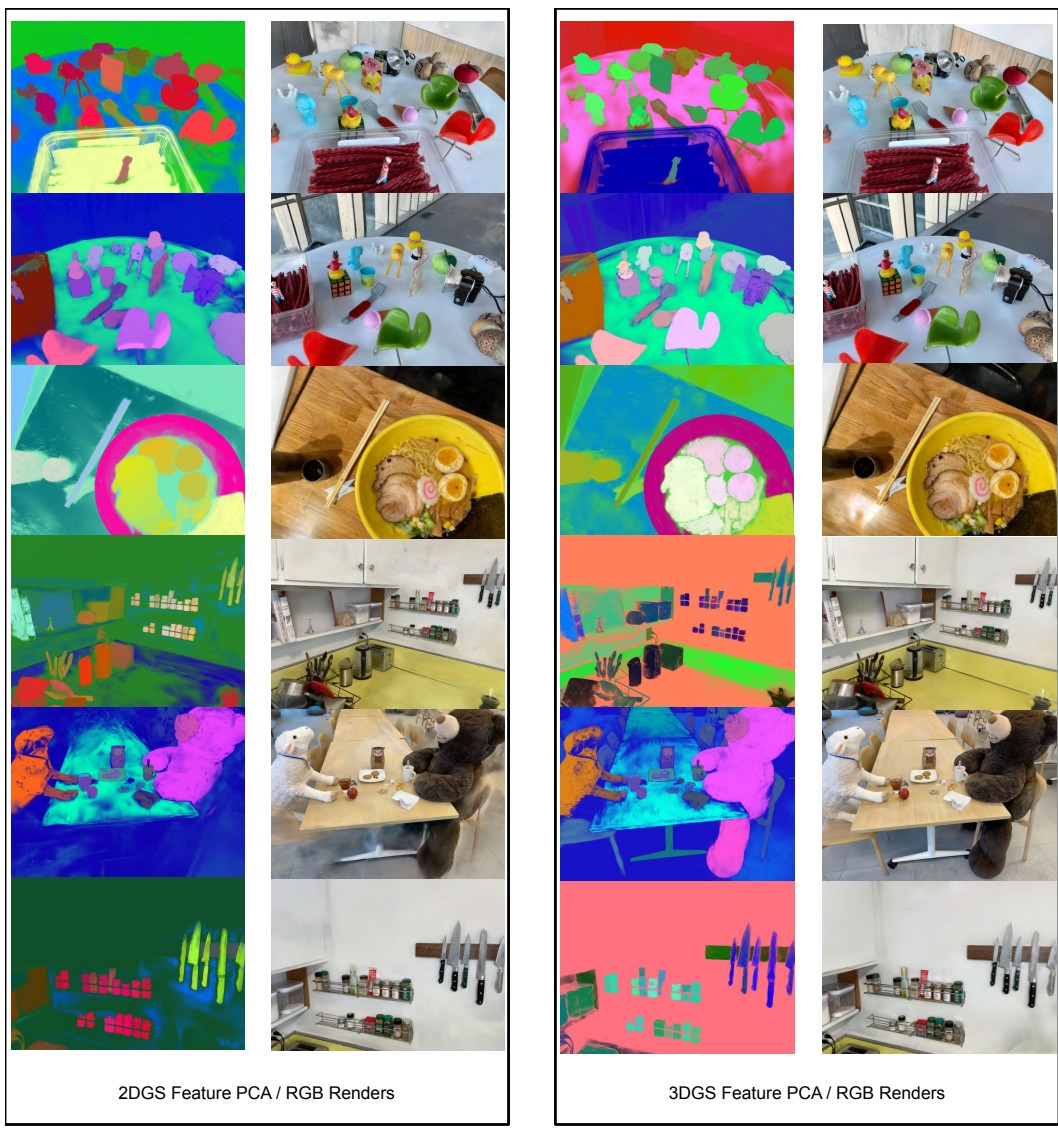

Figure 8: The left column shows the results obtained using Gsplat with 2D Gaussian Splatting (2DGS) for feature lifting, while the right column shows the results using Gsplat with 3D Gaussian Splatting (3DGS). It appears that better reconstruction quality leads to improved PCA outcomes. The lifted features are based on SAM-OpenCLIP embeddings.

## C    Denoising Properties of Post-Lifting Aggregation

The Post-Lifting Aggregation module is a critical component for denoising input errors. It is effective not only at resolving inconsistent segmentation masks but also at mitigating errors introduced by noisy camera poses.

### Implementation

As described in the main paper, we first cluster the initial lifted features using HDBSCAN. Next, we project each 3D cluster onto all training image frustums to identify reliable 2D masks that overlap with the cluster. Masks with an Intersection-over-Union (IoU) overlap above 75% are deemed reliable, while others are discarded during feature aggregation. Finally, we aggregate features only from these reliable masks—merging them into a single feature representation—and assign this unified feature to all splats within the corresponding cluster.

### Intuition

As noted in the "Tikhonov Guidance" section (Appendix B), small noise in observations—manifesting as nearly identical rows in the system matrix—can lead to large errors. A straightforward remedy is to identify and discard these unreliable observations.

Why is it safe to discard them? Recall our problem definition: "each primitive should admit a single descriptor." Theoretically, nearly identical rows (rays hitting the same primitive) should yield nearly identical features. Discrepancies imply errors either from the solver or corrupted observations. Since our queries rely on cluster-level attention scores, noise that affects a cluster uniformly (cluster-independent noise) merely shifts the score by a constant without altering the relative ordering.

In contrast, cluster-dependent noise—often introduced by occlusions or semantic bleed from nearby objects (see Fig. 7)—distorts these scores. By generating a pseudo-mask for each cluster and computing its IoU against the original input masks, we can effectively filter out clusters where the pseudo-mask indicates unreliability.

Crucially, this mechanism makes our method extremely robust to noisy camera poses. Misaligned poses result in low overlap between the projected cluster and the 2D mask, causing those corrupted observations to be automatically filtered out. We validate this robustness in the following experiments.

### Experiments

We first demonstrate robustness against inconsistent masks in Tab. 1 and Tab. 4. Our aggregation module yields a direct gain of 1.9% mIoU even with accurate camera poses, simply by resolving mask inconsistencies.

To evaluate robustness against camera pose noise, we conducted a controlled experiment where we artificially corrupted the poses of a random subset ($N$) of every 100 images in the training set. The splat geometry was fixed (trained on clean poses). The noise injection was defined as follows:

- **Rotation:** We added Gaussian noise with $\sigma = 1°$ to roll, pitch, and yaw.
- **Translation:** We added Gaussian noise with $\sigma = \frac{\text{scene scale}}{100}$.

We then applied our solver to this noisy data. The quantitative results, presented in Tab. 3, demonstrate that our method maintains high performance even when a significant fraction of poses are corrupted. Additionally, we provide a qualitative comparison between results obtained from clean poses versus those with 50% pose corruption in Fig. 9.

## D    Training Pipeline Setting

Achieving high-quality reconstruction is not the primary focus of our study. While our code includes components for training, the reconstruction results are less accurate compared to those obtained

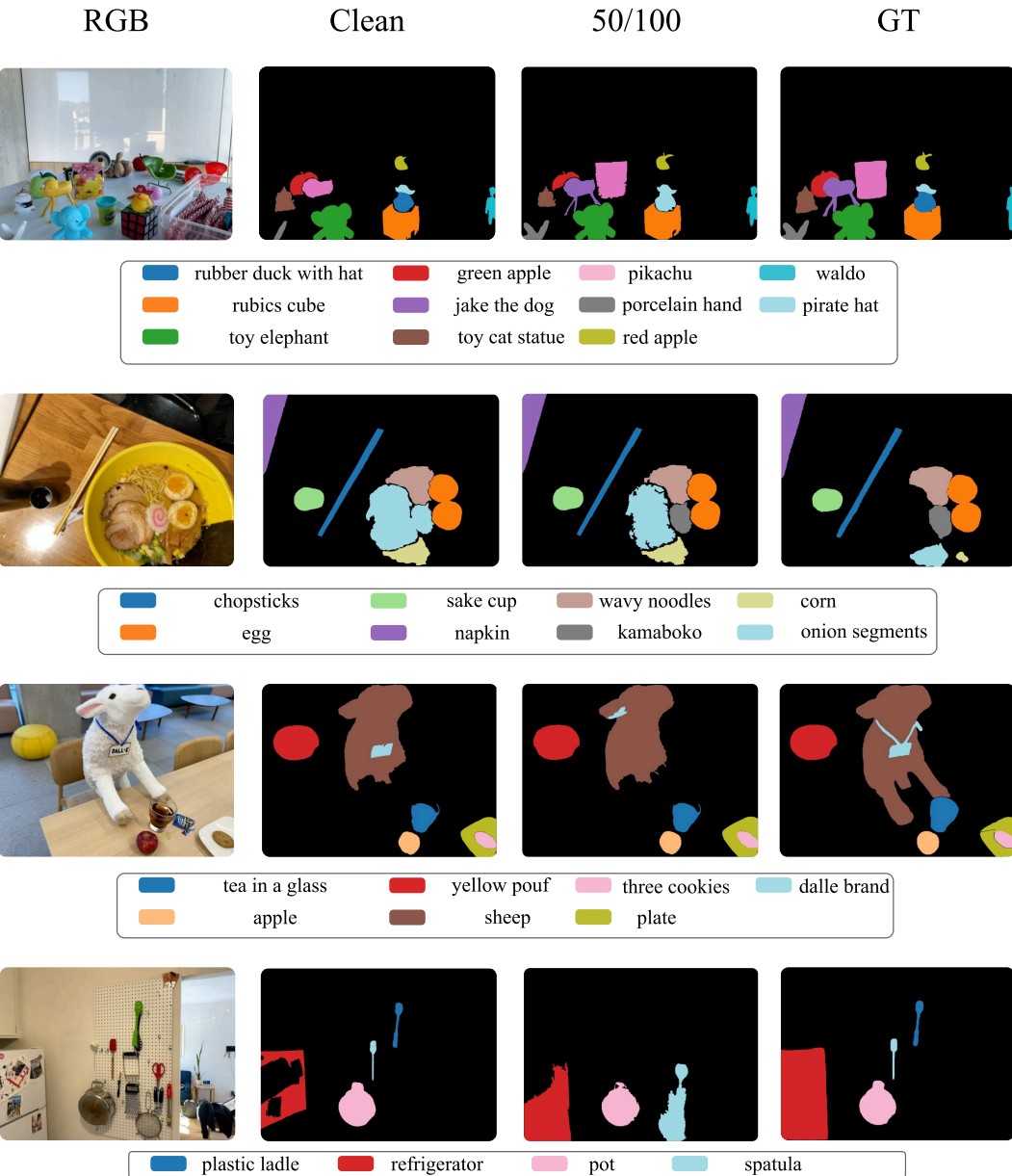

Figure 9: **Qualitative robustness against noisy camera poses.** The columns display (from left to right): the input RGB images, semantic segmentation results using clean camera poses, results with 50% of camera poses corrupted (50/100), and the Ground Truth (GT) segmentation masks. Although there is a slight drop in visual quality, our method maintains consistent segmentation performance even under severe pose noise.

Table 3: **Robustness against noisy camera poses.** The first column indicates the number of images corrupted (per 100 images). The remaining columns report the downstream mIoU scores. Note that cosine similarity is not used here because corrupted camera poses invalidate the ground truth projection alignment. Our method maintains high performance ($\approx 65\%$ mIoU) even when up to 20% of camera poses are noisy, demonstrating significant robustness.

| Corrupted Images (per 100) | Figurines | Ramen | Teatime | Waldo Kitchen | Mean |
|---|---|---|---|---|---|
| 100/100 | 1.43 | 56.0 | 18.1 | 36.8 | 28.1 |
| 50/100 | 64.7 | **71.7** | 58.9 | 48.1 | 60.8 |
| 20/100 | **72.1** | 63.3 | **69.1** | 55.2 | 65.0 |
| 10/100 | 68.8 | 64.8 | 63.9 | 57.4 | 63.7 |
| 3/100 | 65.6 | 64.2 | 63.4 | 59.5 | 63.2 |
| 2/100 | 66.9 | 64.2 | 63.4 | 59.1 | 63.4 |
| 1/100 | 66.2 | 63.0 | 64.7 | 59.8 | 63.6 |
| 0/100 (Clean) | 67.6 | 68.5 | 62.3 | **62.1** | **65.1** |

Table 4: Mean IoU (mIoU%) on LeRF OVS for various methods. These are additional experimental results using our data for the ablation study. We use different backbones to train the splats and evaluate the resulting mIoUs

| Method | Figurines | Ramen | Teatime | Waldo Kitchen | Means |
|---|---|---|---|---|---|
| DrSplat (3D Query) | 47.48 | 36.66 | 66.16 | 47.48 | 49.45 |
| Gsplat (3DGS) W/o P W/o T (Threshold = 0.65) | 55.30 | 47.81 | 63.48 | 49.78 | 54.09 |
| Gsplat 2DGS Backbone W/o P W/o (Threshold = 0.65) | 62.00 | 56.04 | 66.34 | 51.07 | 58.86 |
| Gsplat (3DGS) W/P w/o T (Threshold = 0.65) | 56.55 | 62.24 | 68.04 | 53.01 | 59.96 |
| Gsplat (2DGS) W/P W/o T (Threshold = 0.65) | 67.83 | 60.20 | 66.96 | 47.44 | 60.62 |
| Inria Trained Result W/o P, Naive (W/o Tikhonov) (Threshold) | 60.06 | 55.30 | 68.33 | 47.73 | 57.85 |
| Inria Trained Result W/o P, (Threshold=0.65) | 61.30 | 54.20 | 67.60 | 45.01 | 57.03 |
| Inria Trained W/p W/ Tikhonov (1.2 / 1.2) | 61.70 | 53.75 | 67.80 | 49.60 | 58.21 |
| Inria Trained Result W/p w/ (Tikhonov square, no sigmoid) | 71.96 | 58.16 | 69.29 | 52.73 | 63.03 |
| Inria Trained Result W/p (Threshold 0.65) + W/T | 64.76 | 61.70 | 71.63 | 54.71 | 63.20 |
| Inria Trained W/p W/o Tikhonov Square, w/o sigmoid | 65.45 | 58.55 | 72.04 | 50.38 | 61.61 |
| Inria Trained W/p W/o (Tikhonov $\lambda$ w/o squeer) W/ hist selection | 72.19 | 64.28 | 66.08 | 56.69 | 64.81 |
| Inria Trained W/o T W/p W/ Dynamic Threshold | 69.42 | 63.89 | 69.53 | 61.55 | 66.10 |
| Inria Trained W/ T W/ P W/ Dynamic Threshold | 67.64 | 62.34 | 68.48 | 62.11 | 65.14 |

Table 5: Effect of Tikhonov Guidance ($\lambda_{\mathrm{square}}/\lambda_{\mathrm{reg}}$) on mIoU (%). $\lambda$ Equ denotes the $\lambda$ parameters used when solving the Feature Lifting Equation. When using a text query to generate the attention map, we apply $\lambda$ Proj to obtain the 2D attention map. The experiments show that setting both $\lambda$ values to 1.2 yields optimal results.

| $\lambda$ Equ / $\lambda$ Proj | Figurines | Ramen | Teatime | Waldo Kitchen | Means |
|---|---|---|---|---|---|
| 0.5 / 1.0 | 59.93 | 53.77 | 66.43 | 41.32 | 55.36 |
| 0.8 / 1.0 | 62.77 | 54.02 | 67.38 | 43.90 | 57.02 |
| 1.0 / 1.0 | 61.30 | 54.20 | 67.60 | 45.01 | 57.03 |
| 1.1 / 1.1 | 61.59 | 53.96 | 68.93 | 47.72 | 58.05 |
| 1.2 / 1.2 | 61.70 | 53.75 | 67.80 | 49.60 | 58.21 |
| 1.35 / 1.35 | 61.80 | 53.44 | 65.56 | 50.36 | 57.79 |
| 1.5 / 1.5 | 61.80 | 53.40 | 65.53 | 50.40 | 57.78 |

Table 6: Our methods generated masks versus LAGA generated mask on 3D-OVS dataset, as we mentioned in Fig.6, room scene is not labeled correctly in ground truth, therefore, we provide two results on room scene. The former one is the result measured on original 3DOVS dataset, and the left one is the fixed results. In both comparison, we are at least the same compare to the current state of the art method.

| Methods | Bed | Bench | Lawn | Room | Sofa | Means |
|---------|-----|-------|------|------|------|-------|
| LAGA | 84.9 | 83.3 | **93.6** | **93.1**/86.2 | **76.5** | **86.3**/84.9 |
| Ours | **90.2** | **93.2** | 89.0 | 85.5/**92.7** | 73.6 | **86.3/87.7** |

Table 7: Weight Summation of each rows statistic property. In the Property.2, we assume that the row summation is 1, and in the main thesis, we state that this property usually holds in the splats rendering. Here is experimental results regarding the final weight summation of each pixel. The number is with a %. The dataset we are using is LeRF

| Method Metrics | Figurines | Ramen | Teatime | Waldo Kitchen | Means |
|----------------|-----------|-------|---------|---------------|-------|
| 3DGS(means) Black Background | 99.49 | 97.92 | 99.40 | 99.86 | 99.17 |
| 3DGS(Std) Black Background | 1.37 | 1.54 | 0.67 | 0.18 | 0.94 |
| 3DGS(means) | 99.74 | 99.85 | 99.83 | 99.91 | 99.17 |
| 3DGS(Std) | 0.01 | 0.04 | 0.02 | 0.05 | 0.04 |
| 2DGS(means) | 99.86 | 99.46 | 99.83 | 99.91 | 99.77 |
| 2DGS(Std) | 0.04 | 0.45 | 0.05 | 0.05 | 0.15 |

Table 8: Weight Summation of each rows statistic property. In the Property.2, we assume that the row summation is 1, and in the main thesis, we state that this property usually holds in the splats rendering. Here is experimental results regarding the final weight summation of each pixel. The number is with a %. The dataset we are using is 3DOVS

| Methods | Bed | Bench | Lawn | Room | Sofa | Means |
|---------|-----|-------|------|------|------|-------|
| 3DGS(means) | 99.92 | 99.80 | 99.67 | 99.64 | 99.93 | 99.79 |
| 3DGS(Std) | 0.01 | 0.04 | 0.08 | 0.04 | 0.01 | 0.04 |
| 2DGS(means) | 99.92 | 99.93 | 99.95 | 99.92 | 99.92 | 99.93 |
| 2DGS(Std) | 0.01 | 0.01 | 0.00 | 0.01 | 0.01 | 0.01 |

using the LAGA training code. Therefore, in the main comparison table, we use LAGA's pre-trained model. For visualization experiments, we adopt the Gsplat implementation for both 2DGS and 3DGS to ensure a fair comparison. As shown in Fig.8, the visualizations suggest that reconstruction quality does influence the feature lifting results.

## E    LIFTING IMPLEMENTATION

Our code is publicly available in the supplementary material and on the project website. We adopt the original rendering pipeline for feature lifting, with a modification to the final alpha-blending step: instead of projecting each splat's color into the 2D image, we back-project 2D feature information into the corresponding 3D splat feature and record the blending weights at runtime. This approach is more memory-efficient than DrSplats, which explicitly constructs pixel-to-splat correspondences. As a result, DrSplats may encounter CUDA out-of-memory (OOM) errors when the maximum depth increases, whereas our method is capable of handling significantly larger inputs. In principle, any 3D image produced by the rendering pipeline can be lifted in a similar way by slicing its rendered features.

## F    3D SEGMENTATION

For 3D segmentation, splats can be selected by comparing attention scores associated with target ('positive') words against those linked to background words. To avoid the complexity of explicit background-word selection, we compute direct attention scores for each splat and render them as one-dimensional features in the 2D image—thereby minimizing background interference. Since the raw attention scores typically fall outside the [0,1] interval, we rescale them for visualization and manual thresholding. However, during the dynamic threshold selection step, we operate directly on the un-normalized attention scores to determine the optimal threshold. Further visualization results can be found in the appended **video**.

## G    PROPERTY JUSTIFICATION

Recall from property.2, this property means for each ray, there should be nearly no influence of the background color. Let us first justify why this property is necessary. If without the above property, even the optimized lift will depend on the background feature. When we utilize random feature background, the error has a lower bound which is the left out alpha for each ray. This justification will also hold true if we are utilizing color as a feature. To justify this property, we first scratch a theoretical proof, and then, we give out experimental results.

THEORETICAL

$$C'_b \sim \mathcal{U}(-1, 1) \tag{19}$$

$$C_r = \sum_p \omega_p c_p + \left(1 - \sum_p \omega_p\right) C'_b \tag{20}$$

$$s = \sum_p \omega_p \quad L(\omega) = MSE(C_r, \hat{C}_r) \tag{21}$$

$$L(\omega) = \mathbb{E}_{C'_b}\left[\left(C_r - \hat{C}_r\right)^2\right] \tag{22}$$

$$= \mathbb{E}_{C'_b}\left[\left(\sum_p \omega_p c_p - \hat{C}_r + (1-s) C'_b\right)^2\right]. \tag{23}$$

Here we define the ground truth for per ray color as $\hat{C}_r$, we use MSE as a loss function to calculate the loss between observed color and rendered color. The goal of the proof is to show that under any gradient descent based method, if the loss converges, $s$ approaches to one. Notice that here we define the background color $C'_b$ as a random background that has a uniform distribution on the normalized

color definition. Let us take the stochastic gradient with respect to weight $\omega$, and the summation of weight $s$.

$$\frac{\partial L}{\partial \omega_p} = 2(C_r - \tilde{C}_r)(C_p - C_b') \tag{24}$$

$$\frac{\partial L}{\partial s} = \sum_p \frac{\partial L}{\partial \omega_p} = 2\left(C_r - \tilde{C}_r\right)\sum_p (C_p - C_b') \tag{25}$$

Here we substitute the residual $\epsilon$ back to the Equ.24:

$$\epsilon = \sum_p \omega_p C_p - \tilde{C}_r, \quad C_r - \tilde{C}_r = \epsilon + (1-s)C_b' \tag{26}$$

$$\mathbb{E}\left[\frac{\partial L}{\partial s}\right] = 2\,\mathbb{E}\Big[\left(\varepsilon + (1-s)\,C_b'\right)\sum_p (C_p - C_b')\Big]. \tag{27}$$

When we expand the above equation, we get the expectation of the gradient is the Equ.28

$$\mathbb{E}\left[\frac{\partial L}{\partial s}\right] = \mathbb{E}\Big[2\,\varepsilon\sum_p c_p - 2\,\varepsilon\sum_p (c_p - C_b') \\ + (1-s)\,C_b'\sum_p c_p + (s-1)\,(C_b')^2\Big]. \tag{28}$$

And apparently, we can get everything canceled out, except this term shown in Equ.29

$$\mathbb{E}\left[\frac{\partial L}{\partial s}\right] = (s-1)\sigma^2 \tag{29}$$

Therefore, to converge, $s - 1$ must be zero, i.e. $\sum_p \omega_p = 1$ Q.E.D. According to Property.3, we usually have a converged system at least on RGB, therefore this property holds.

### EXPERIMENTAL

To experimentally validate our theoretical prediction, we conducted the following experiment. We trained both 3D Gaussian Splats and 2D Gaussian Splats separately using random backgrounds, then computed the mean and variance of their final alpha distributions. If the mean exceeds a threshold $\psi$ = 99.6, our theory is confirmed. We selected $\psi$ = 99.6 because color values are quantized into 256 levels, and additional floating-point precision does not improve RGB rendering. For comparison, we also ran the 3D Gaussian Splats experiment with a black background. As shown in Tables 7 and 8, the results align well with our theoretical proof.

$$\psi = (1 - \frac{1}{\text{Quantize Scale}}) \times 100 = 99.6 \tag{30}$$

## H  FEATURE AGNOSTIC AND KERNEL AGNOSTIC

### FEATURE AGNOSTICISM

Recall from Section 3.2 that our solver is proven to be optimal under any convex loss function. Theoretically, this guarantees a tighter error bound compared to heuristic or optimization-based approaches such as Chacko et al. (2025); Jun-Seong et al. (2025); Guo et al. (2024); Joseph et al. (2024). While Table 1 demonstrated that our method consistently achieves a cosine similarity above 80%, that result alone did not quantify our superiority over specific baselines.

To address this, we provide a detailed comparison against current feature-agnostic methods, specifically Guo et al. (2024); Chacko et al. (2025). We exclude Dr.Splat Jun-Seong et al. (2025) from this comparison because its pipeline necessitates compressing original features (e.g., 512 dimensions) down to low-dimensional subspaces (e.g., 32 dimensions), thereby losing feature generality. Similarly, grouping-based methods are excluded as they cannot operate without mask guidance. As

RGB       Rendered       Observation

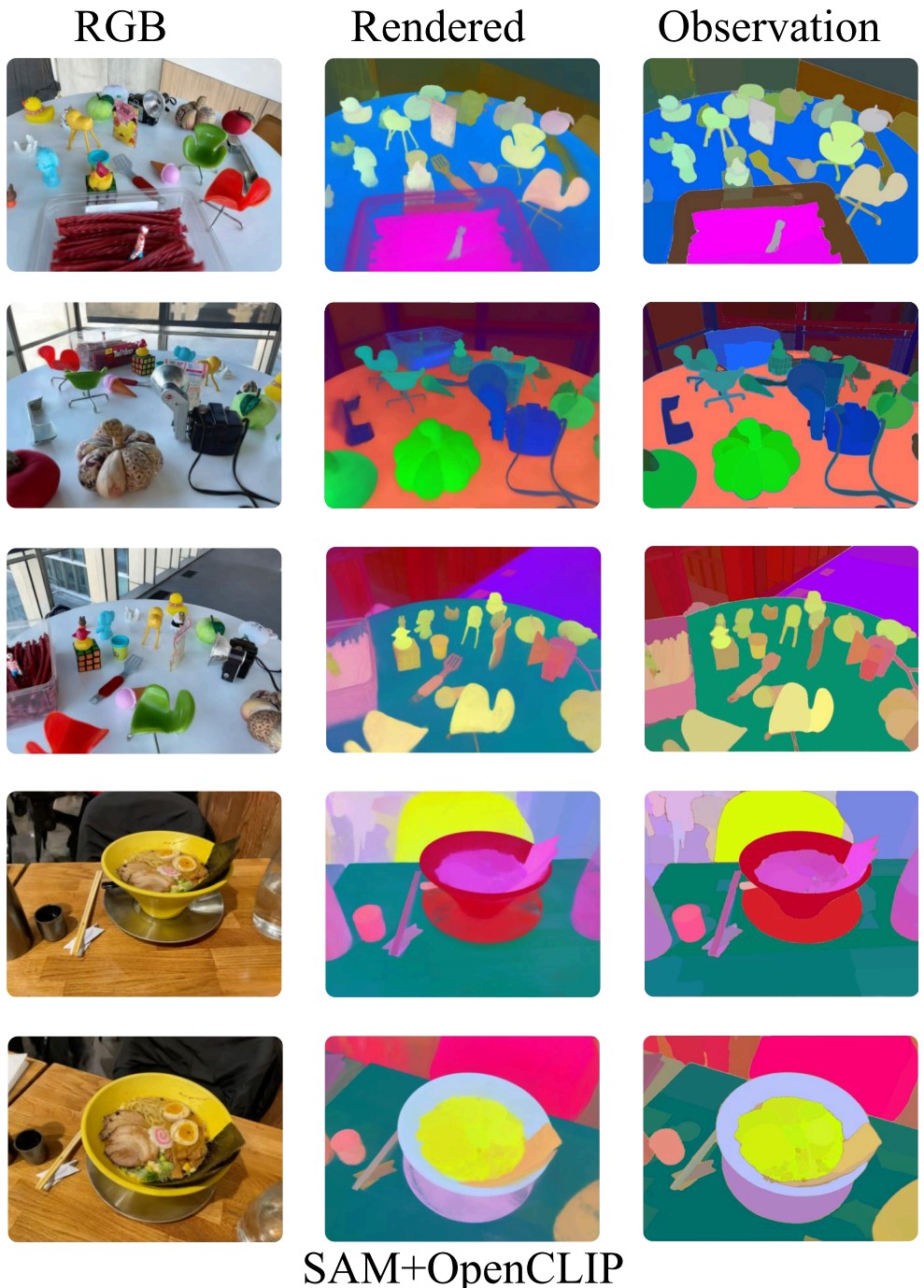

SAM+OpenCLIP

Figure 10: **Feature Map Comparison:** This figure compares our rendered features against the ground truth 2D features. The ground truth visualization is generated by applying PCA to the feature maps extracted directly from the input image by the foundation model. The rendered features represent the PCA of the features projected from our solved splats. These results demonstrate the SAM+OpenCLIP pipeline. Additionally, we compare our lifted result to current SOTA lifting method in Fig.13

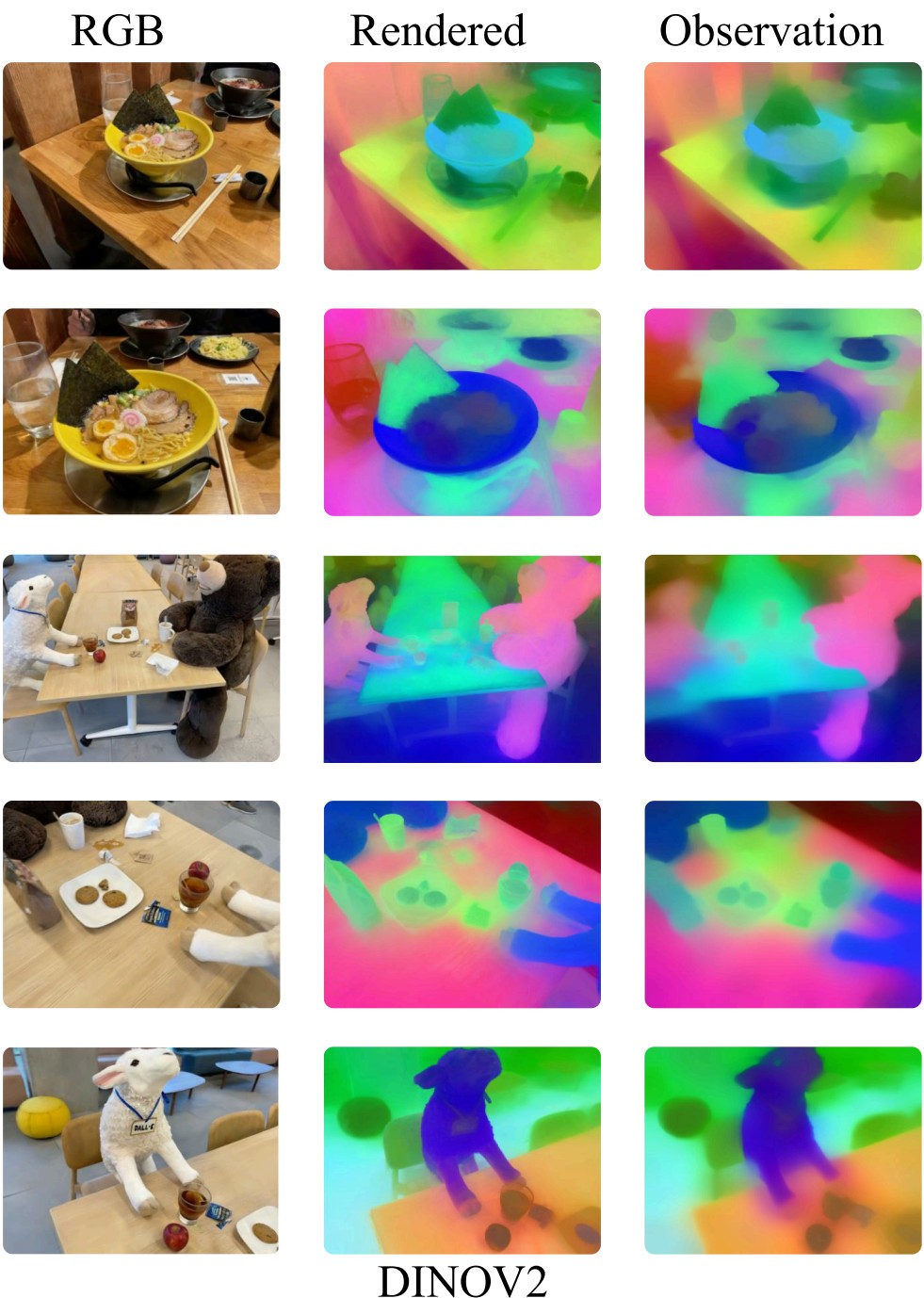

Figure 11: **Feature Map Comparison:** This figure compares our rendered features against the ground truth 2D features. The ground truth visualization is generated by applying PCA to the feature maps extracted directly from the input image by the foundation model. The rendered features represent the PCA of the features projected from our solved splats. These results demonstrate the DINOv2 pipeline. Additionally, we compare our lifted result to current SOTA lifting method in Fig.14

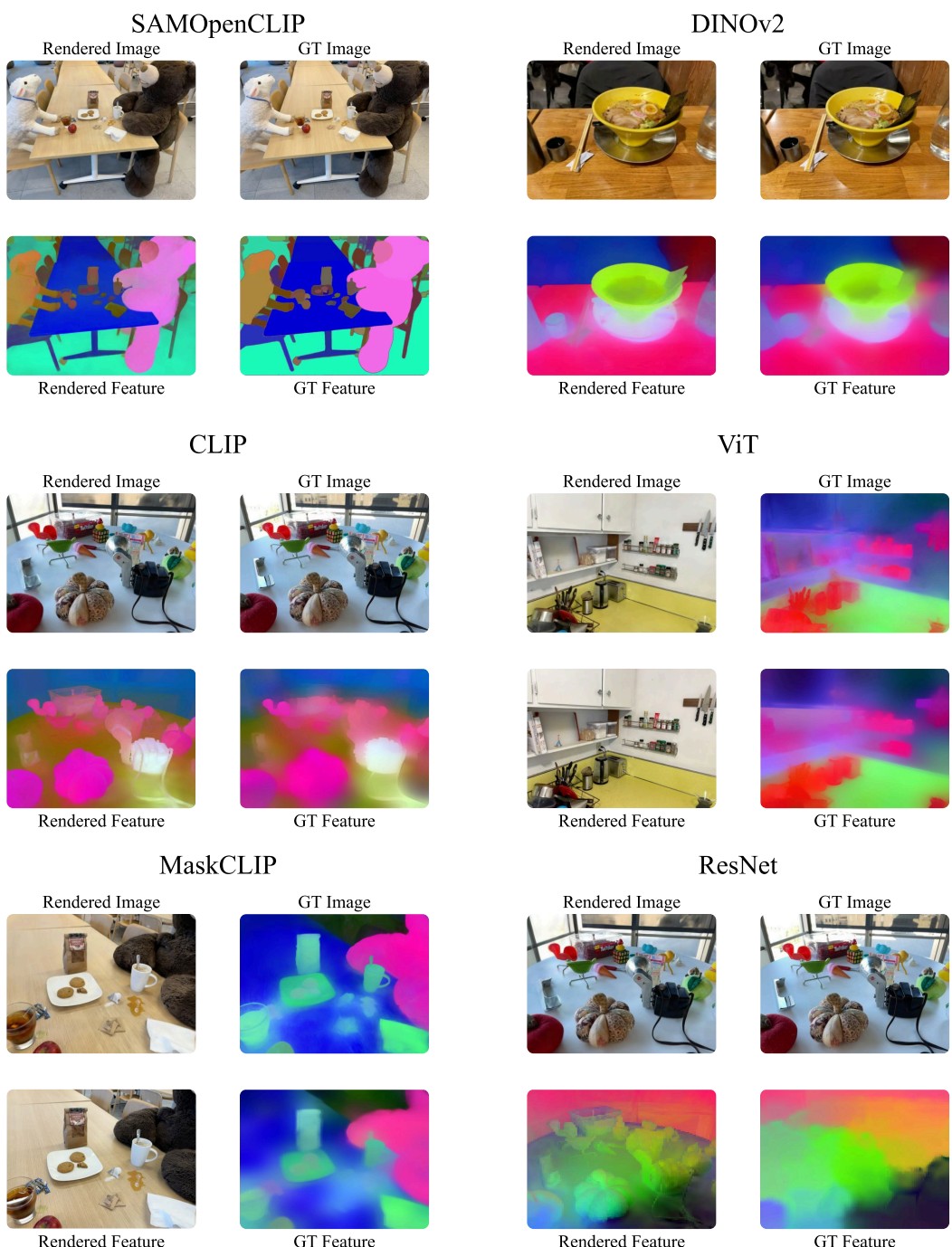

Figure 12: Further Visualization results, we are using Teatime, ramen, figurines, waldo kitchen results from different scenes. More visualization results could be found in the supplementary materials

Ours            Semantic GS            ArgMaxLifting

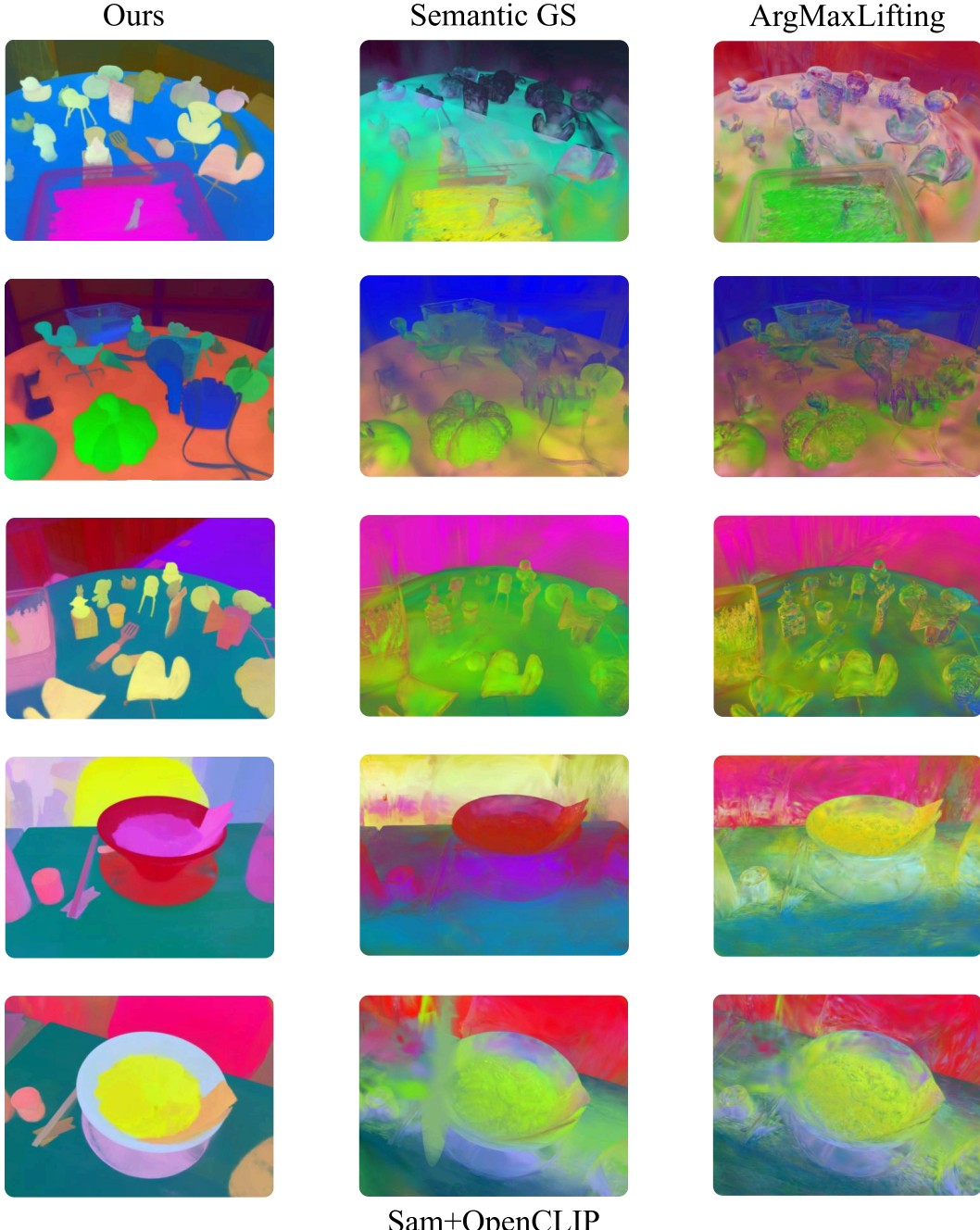

Sam+OpenCLIP

Figure 13: **Qualitative Comparison of Feature Fidelity:** We visualize the principal components (PCA) of the lifted feature fields rendered into 2D using the SAM+OpenCLIP backbone. We compare our proposed method (left columns) against another optimization-based Semantic Gaussian Guo et al. (2024) (middle column) and the heuristic Argmax Lifting Chacko et al. (2025) (right column). The input RGB and ground truth feature map are the same as the Fig.10. Our method recovers sharper object boundaries and more consistent internal feature representations (higher fidelity), avoiding the floating Gaussian artifacts and the noise inherent in heuristic aggregation.

Ours                          Semantic GS                   ArgMaxLifting

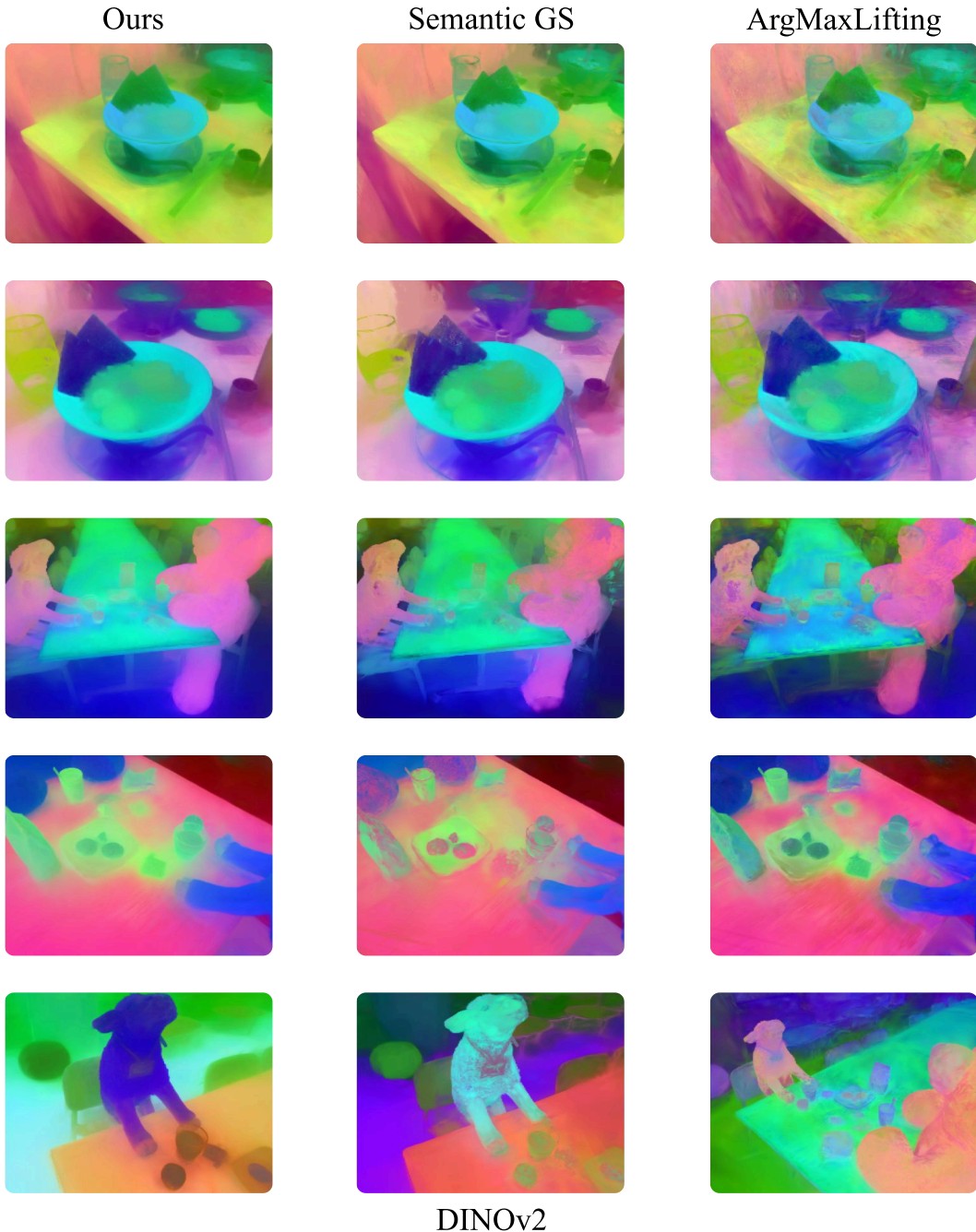

DINOv2

Figure 14: **Qualitative Comparison of Feature Fidelity:** We visualize the principal components (PCA) of the lifted feature fields rendered into 2D using the DINOv2 backbone. We compare our proposed method (left columns) against another optimization-based Semantic Gaussian Guo et al. (2024) (middle column) and the heuristic Argmax Lifting Chacko et al. (2025) (right column). The input RGB and ground truth feature map are the same as the Fig.11. Our method recovers sharper object boundaries and more consistent internal feature representations (higher fidelity), avoiding the floating Gaussian artifacts and the noise inherent in heuristic aggregation.

Table 9: Comparison of feature lifting fidelity (Cosine Similarity). Red indicates the best performance (highest), Pink indicates the second best, and Green indicates the lowest performance. Our method consistently achieves the highest fidelity across all feature types.

| Feature | Method | F. | T | R | W. | Mean |
|---|---|---|---|---|---|---|
| SAM+OpenCLIP Kirillov et al. (2023) | Semantic Gaussian Guo et al. (2024) | 80.7 | 83.1 | 83.7 | 85.2 | 83.2 |
| | Argmax Lifting Chacko et al. (2025) | 81.8 | 82.9 | 83.8 | 85.4 | 83.4 |
| | Ours | 89.3 | 90.7 | 90.6 | 91.0 | 90.4 |
| MaskCLIP Dong et al. (2023) | Semantic Gaussian Guo et al. (2024) | 90.6 | 93.0 | 94.0 | 93.1 | 92.8 |
| | Argmax Lifting Chacko et al. (2025) | 91.3 | 92.3 | 93.3 | 94.2 | 92.8 |
| | Ours | 92.4 | 94.0 | 94.5 | 94.7 | 93.9 |
| CLIP Radford et al. (2021) | Semantic Gaussian Guo et al. (2024) | 90.4 | 92.8 | 94.1 | 93.7 | 92.8 |
| | Argmax Lifting Chacko et al. (2025) | 90.8 | 91.6 | 93.4 | 94.3 | 92.5 |
| | Ours | 91.9 | 93.7 | 94.6 | 94.9 | 93.8 |
| DINO Caron et al. (2021) | Semantic Gaussian Guo et al. (2024) | 72.5 | 76.5 | 81.3 | 75.2 | 76.4 |
| | Argmax Lifting Chacko et al. (2025) | 74.0 | 72.3 | 78.3 | 78.9 | 75.9 |
| | Ours | 78.7 | 80.2 | 83.0 | 82.0 | 81.0 |
| DINOv2 Oquab et al. (2023) | Semantic Gaussian Guo et al. (2024) | 77.5 | 82.5 | 88.2 | 83.9 | 83.0 |
| | Argmax Lifting Chacko et al. (2025) | 80.2 | 79.6 | 87.2 | 87.3 | 83.6 |
| | Ours | 83.5 | 85.6 | 89.6 | 89.2 | 87.0 |
| ViT Dosovitskiy et al. (2020) | Semantic Gaussian Guo et al. (2024) | 81.7 | 82.6 | 85.8 | 85.4 | 83.9 |
| | Argmax Lifting Chacko et al. (2025) | 81.6 | 77.8 | 83.7 | 86.2 | 82.3 |
| | Ours | 84.6 | 83.7 | 86.4 | 88.0 | 85.7 |
| ResNet He et al. (2016) | Semantic Gaussian Guo et al. (2024) | 94.9 | 94.4 | 96.0 | 94.4 | 94.9 |
| | Argmax Lifting Chacko et al. (2025) | 94.8 | 93.1 | 95.1 | 96.5 | 94.9 |
| | Ours | 95.8 | 94.8 | 96.2 | 97.0 | 96.0 |

shown in Table 9, the high similarity range (80–90%) achieved by our method represents a significant improvement over these existing approaches.

For qualitative feature visualizations, please refer to Fig. 10, Fig. 11, and Fig. 12. Comparisons with existing feature-agnostic methods, such as Chacko et al. (2025); Guo et al. (2024), are provided in Fig. 13 and Fig. 14.

KERNEL AGNOSTICISM

For multiple kernel feature visualization, one can check Fig.8. There are plenty of the content in the appended **video**.

# I THRESHOLD

We can refer directly to Fig. 15. The basic idea is to trace the gradient from top to bottom, then use the first local maximum and the preceding local minimum to determine the threshold. In this way, the threshold adapts dynamically and is independent of the background words.

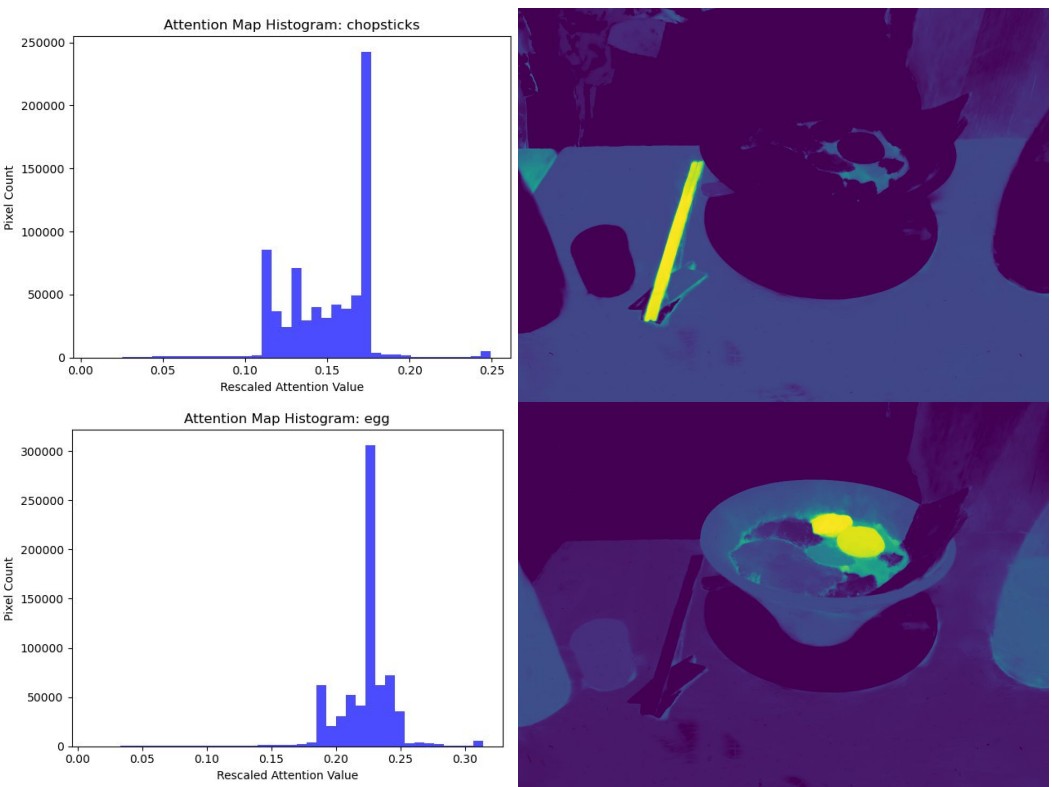

Figure 15: This figure is from the Ramen scene in the LeRF dataset. The query word is "chopsticks" for the first row, and "eggs" for the second row. The first column represents the attention maps' histograms while the second column is the actual attention map. A simple gradient-based dynamic threshold selection strategy can be readily envisioned for downstream tasks such as 3D segmentation.

