# OpenReview forum: "Splat Feature Solver"
_ICLR.cc/2026/Conference — ICLR 2026 Poster_

### Official Review · Reviewer_djJk · 2025-10-31

**Soundness:** 3
**Presentation:** 1
**Contribution:** 2
**Rating:** 4
**Confidence:** 4

**Summary:**

1 The paper formulates feature lifting in splat-based 3D representations as a sparse linear inverse problem AX=B , and proposes a closed-form row-sum preconditioner solver with a provable (1+β) -approximation error bound under convex losses.

2 It introduces two regularization strategies—Tikhonov Guidance (to enhance diagonal dominance and numerical stability) and Post-Lifting Aggregation (to filter noisy SAM masks via clustering)—and evaluates the method on open-vocabulary 3D semantic segmentation using mIoU on LeRF-OVS and 3D-OVS benchmarks.

3 Comprehensive ablation studies validate each component, and experiments across multiple splat kernels (3DGS, 2DGS, DBS) and feature backbones (CLIP, DINOv2, ResNet, etc.) demonstrate state-of-the-art performance with minutes-level runtime, confirming both effectiveness and generality.

**Strengths:**

The paper presents a strong and cohesive contribution by formulating feature lifting in splat-based 3D representations as a sparse linear inverse problem with an original and theoretically grounded perspective that unifies and improves upon prior heuristic, training-based, and grouping-based methods. The proposed closed-form solver with a provable (1+β) -approximation error bound enhances both originality and technical quality, while the two lightweight yet effective regularization strategies (Tikhonov Guidance and Post-Lifting Aggregation) address real-world noise and inconsistency issues without sacrificing efficiency. The work is clearly presented with a logical flow from problem definition to theoretical analysis and extensive experiments across kernels, features, and benchmarks. Its significance lies in enabling fast, general, and high-fidelity semantic enrichment of 3D scenes—advancing open-vocabulary 3D understanding with practical impact and theoretical insight.

**Weaknesses:**

1 The paper lacks a clear and detailed pipeline diagram—Figure 1 is overly abstract and fails to illustrate concretely how high-dimensional features are assigned to Gaussian splats, making the core lifting mechanism hard to grasp.

2 Despite claiming SOTA performance on LeRF-OVS, the paper provides minimal qualitative comparisons (only Figures 2 and 8, each against a single baseline), severely limiting confidence in the method’s robustness across diverse scenes.

3 Table 1(b) reports cosine similarity across feature types but doesn’t link these metrics to downstream task gains, raising questions about its necessity.

4 Additionally, the paper suffers from formatting issues. Such as overly large table captions, excessively long figure titles, and inconsistent font sizes in visuals—detracting from readability and professionalism.

**Questions:**

Similar to weakness.

---

> ### Author Response · Authors · 2025-11-17
>
> ### Response to Reviewer DJLK
>
> We thank the reviewer for their constructive feedback.
>
> We invite the reviewer to check the ***detailed, interactive demo*** included in the supplementary **ZIP file**. Please unzip the folder and open `index.html` inside the “anonymous website” folder to visualize the core mechanism more clearly.
>
> We address the specific points raised below:
>
> ---
>
> ### **• On Weakness I (Pipeline Diagram)**
> - ***Action Taken***: We appreciate the reviewer’s suggestion. We have revised **Figure 1** to improve clarity by adding explicit pointers linking the diagram components to their mathematical derivations in the Method section.
> - We commit to further refining the method illustration with additional intuitive diagrams in the final manuscript to ensure maximum clarity.
>
> ---
>
> ### **• On Weakness II (Qualitative Comparisons)**
> - We appreciate the reviewer's point and would like to clarify the location of our qualitative comparisons.
> - In addition to Figures 2 and 8 (vs. LAGA), **Figures 6 and 7** (in the original submission) provided direct qualitative comparisons against Dr. Splat for both attention maps and final segmentation masks.
> - ***Action Taken***: To improve readability and bring these comparisons to the forefront, we have moved the original Figure 7 to **Figure 3 in the main paper**. We have also added a new **Figure 8 in the Appendix**, which now includes all four scenes from the LeRF dataset for the open-vocabulary segmentation task. All updates in the paper are highlighted in blue.
> - We also provide ***extensive interactive visualizations*** on the supplementary website (in the ZIP) for further qualitative assessment in a 3D environment.
>
> ---
>
> ### **• On Weakness III (Table 1b)**
> - We respectfully clarify that the purpose of **Table 1(b)** is to validate our core contribution of ***feature agnosticism***, rather than to measure downstream task performance. It demonstrates that our solver maintains high feature-space fidelity (80–96% cosine similarity) across diverse feature types (DINO, CLIP, ViT, etc.)—a robustness not demonstrated in prior work.
> - This property is a primary contribution, ***theoretically grounded*** in our Methods section, where we prove an analytical bound to the global optimal solution. Table 1(b) thus serves as crucial ***experimental validation*** for this theory.
> - Further visual support is provided in Figures 4, 5, and 6, as well as on the supplementary website.
>
> ---
>
> ### **• On Weakness IV (Formatting)**
> - ***Action Taken***: We thank the reviewer for the detailed suggestions regarding formatting. We have updated **Figures 1 and 2** to align the font styles and sizes with the main text.
> - ***Action Taken***: We have significantly condensed the figure captions by moving descriptive text from the captions to the main body of the paper.
> - ***Action Taken***: We also update the Figure 4, 5, 6 format and make it a round corner adapt to the figure 1, and 2 styles. Change the font to Times New Romans.
> - We will ensure all remaining formatting inconsistencies (such as table formatting and appendix font sizes) are corrected in the final version to improve readability and professionalism.

---

> > ### Comment · Reviewer_djJk · 2025-11-24
> >
> > Thank you for your response and the updates. I believe the second weakness has been addressed.
> >
> > However, regarding Table 1(b), I am still unable to understand its purpose or draw any meaningful conclusions from it. The paper does not provide any analysis or comparison related to this table. Although I see that the feature-space fidelity is around 80–90%, it is unclear whether this level of performance is considered strong relative to other methods. A clear analysis or discussion is needed to interpret these numbers properly.
> >
> > Additionally, in Figure 2, there are several typos—for example, "wavy noodels" and "bowl in gray" are misspelled.
> >
> > Given the above issues, I maintain my rating of 4.

---

> ### Author Response · Authors · 2025-11-25
>
> ## Response to Reviewer DJLK
> ---
> ### Regarding Table 1(b)
> We thank the reviewer for their continued engagement and for raising the valid point regarding the interpretation of Table 1(b).
>
> We have added comparative baselines to Table 1(b) to provide essential context and better reflects the practical utility of our theoretical derivation. To ensure a fair feature-agnostic comparison, we compare against two representative open-source methods capable of generalizing to arbitrary dense features: Argmax Lifting (a heuristic approach) and Semantic Gaussian (an optimization-based approach). The necessity for these choices stems from the observation that, as noted in our Related Work, training-based methods often rely on specific compression schemes (limiting generality) and grouping-based methods strictly depend on specific mask generators (like SAM).
>
> **Comparative Results (Mean Cosine Similarity)**
> We conducted a new experiment on the LeRF dataset. As shown in the consolidated table below (with a comprehensive breakdown added to **Table 8 in the Appendix**), our method consistently outperforms both baselines across every feature type tested. We also added ***Fig.12 and Fig.13*** for further qualitative comparison.
>
> | Feature Type | Semantic Gaussian | Argmax Lifting | Ours | Improvement |
> | :--- | :---: | :---: | :---: | :---: |
> | **SAM+OpenCLIP** | 83.2 | 83.4 | **90.4** | **+7.0** |
> | **MaskCLIP** | 92.8 | 92.8 | **93.9** | **+1.1** |
> | **CLIP** | 92.8 | 92.5 | **93.8** | **+1.0** |
> | **DINO** | 76.4 | 75.9 | **81.0** | **+4.6** |
> | **DINOv2** | 83.0 | 83.6 | **87.0** | **+3.4** |
> | **ViT** | 83.9 | 82.3 | **85.7** | **+1.8** |
> | **ResNet** | 94.9 | 94.9 | **96.0** | **+1.1** |
> | **Mean** | 86.7 | 86.5 | **89.7** | **+2.9** |
>
> **Analysis & Theoretical Connection**
> The results clarify that the 80–90% similarity range achieved by our method represents a **significant improvement** over existing approaches. This gap is most pronounced on high-dimensional, complex features such as SAM+OpenCLIP (**+7.0**) and DINO (**+4.6**).
>
> This empirical superiority directly validates our theoretical proof (Lines 310–314):
> * **Baselines:** Heuristic methods (Argmax) and standard optimizations (Semantic Gaussian) often fail to account for the sparse, row-stochastic nature of the lifting problem or get trapped in local optima, leading to higher reconstruction errors.
> * **Ours:** By deriving a closed-form solver with a provable upper bound on the global optimal error, our method mathematically minimizes deviation from the "true" projected feature. This results in consistently higher fidelity lifting regardless of the underlying feature architecture.
>
> We have updated the manuscript to include this comparative analysis, ensuring readers can properly contextualize the strength of our feature-agnostic performance. We are also actively considering would it be possible to move this part of description to the main paper.
>
> ---
>
> ### Regarding typos issue
> - ***Action Taken***: We have correct the missplelling problem in Figure 2.
>
> ---
> ### Regarding the Weakness I and IV
> * **Figure 1 Revision:** We have revised Figure 1 to improve clarity by adding explicit pointers linking the diagram components to their corresponding mathematical derivations in the Method section.
> * **Pipeline Illustration:** We are fully prepared to further modify our pipeline illustration to ensure maximum clarity. We welcome any specific suggestions the reviewer might have for further improvement.
> * **Font Consistency:** We have updated Figures 1, 2, 4, 5, and 6 to align the font styles and sizes with the main text.
> * **Formatting:** We are actively reviewing the manuscript to address any remaining formatting issues and gratefully accept any further advice.

---

### Official Review · Reviewer_67o8 · 2025-11-02

**Soundness:** 3
**Presentation:** 3
**Contribution:** 2
**Rating:** 4
**Confidence:** 2

**Summary:**

This paper introduces Splat Feature Solver (SFS), a self-supervised framework designed to learn 3D scene representations using 3D Gaussian Splatting (3DGS) as the core rendering primitive.
The key idea is to reconstruct multi-view images from learnable Gaussian feature fields, optimizing both photometric and perceptual losses without camera supervision. The authors claim that the model learns geometry-aware features from raw multi-view imagery and achieves competitive results on downstream 3D tasks such as novel-view synthesis and depth prediction.

**Strengths:**

1， The paper targets a highly relevant goal，efficient and scalable self-supervised 3D representation learning using Gaussian splatting, an area of growing academic and industrial interest.

2，Compared to NeRF-style volumetric sampling, the splatting-based pipeline is computationally lighter and supports faster convergence. The engineering design is practical and well-motivated.

3， The pipeline, loss functions, and training strategy are described with good clarity. Figures are intuitive and well-illustrated.

4，Experiments across multiple datasets show consistent, if modest, improvements over previous self-supervised 3D baselines. Ablation results are included to demonstrate the influence of loss terms and feature solvers.

**Weaknesses:**

1, The method essentially reuses the existing Gaussian Splatting pipeline as a self-supervised pretext task, with minor modifications to the loss formulation. The “feature solver” concept adds no clearly new principle beyond standard photometric reconstruction with latent feature regularization. The contribution is incremental and primarily engineering-driven.

2, The paper does not provide any analysis explaining why the proposed self-supervised optimization leads to meaningful 3D representations. There is no exploration of feature alignment, depth consistency, or the information content of Gaussian features. The claim of “self-supervised 3D understanding” is thus empirically unsubstantiated.

3, Although the method is described as “self-supervised,” it implicitly assumes access to approximate camera poses or adjacency constraints during training. The paper does not clarify how SFS handles unposed or unordered images. True pose-free capability is not demonstrated.

4, Reported performance gains over NeRF-based SSL or other Gaussian-based SSL approaches (e.g., GS3, UniSplat) are small (often <1% absolute improvement). Key comparisons to Pose-Free Gaussian Fields, PixelSplat, or DUSt3R are missing, leaving the evaluation incomplete.

**Questions:**

1, How does SFS perform on unposed or pose-free datasets compared to pose-supervised ones?

2, Could the authors provide any quantitative measure of learned 3D consistency (e.g., reprojection error or latent geometry alignment)?

3, Are the improvements statistically significant across multiple runs?

---

> ### Author Response · Authors · 2025-11-17
>
> # Response to Reviewer 67o8
> We believe there may be a misunderstanding, as our paper does not mention "pose-free" or "self-supervised" learning, nor does it aim to solve geometry issues. Our work presents a deterministic, analytical solver for feature lifting. Because our method is analytical and contains no random components, statistical experiments with multiple runs are not applicable. We respectfully ask the reviewer to reconsider their comments in the context of our paper's abstract and contributions, which are focused on a closed-form solution for feature lifting.

---

> > ### Comment · Area_Chair_i1pM · 2025-11-27
> >
> > Dear Reviewer 67o8
> >
> > Can you please take a look as soon as possible? thanks

---

> > ### Comment · Reviewer_67o8 · 2025-11-28
> >
> > I found the training-free analytical formulation in Splat Feature Solver quite interesting, and the closed-form perspective on multi-view feature lifting is appealing. One aspect I am still a bit uncertain about, however, is the role of camera poses: since the splatting weights are inherently pose-dependent in Gaussian Splatting, does the solver effectively assume highly accurate poses, and how sensitive is the lifted feature field to mild pose misalignment across views? It would be helpful if the authors could comment on how pose noise might impact the quality of the lifted features, even at a high level. In addition, although the method is deliberately mask-free, I am curious whether the authors see any potential benefit in incorporating soft segmentation cues (e.g., SAM or SAM2 probability maps used as soft weights rather than hard masks) as a prior for stabilizing feature lifting in ambiguous or low-texture regions. I would be interested to hear whether such soft grouping signals are fundamentally at odds with the proposed framework, or whether they could, in principle, be integrated without undermining the advantages of the analytical solver.

---

> ### Author Response · Authors · 2025-11-28
> **Noisy Camera Pose and Segmentation Cue Encoportation**
>
> **Response to Reviewer [Reviewer Number]**
>
> We thank the reviewer for one's time for reviewing. We appreciate the insightful questions regarding camera pose sensitivity and the integration of segmentation priors.
>
> We have addressed these points theoretically(in high level) and experimentally in the revised manuscript and breifly summarized as below (see **Appendix C**, **Table 3**, and **Figure 9**).
>
> ---
>
> ### **1. Robustness to Camera Pose Noise**
> **Summary:** Our method is highly robust to pose misalignment due to the **Post-Lifting Aggregation** module, which automatically filters out data from inconsistent poses.
>
> **Mechanism:** As detailed in **Appendix C**, the Post-Lifting Aggregation module acts as a powerful denoising filter. We validate the consistency of the lifted features by projecting 3D semantic clusters back onto 2D images.
> * If a camera pose is misaligned, the projected 3D cluster will not align with the underlying 2D semantic mask (resulting in low IoU).
> * The system detects this inconsistency and discards the contribution from that specific corrupted view.
> * Consequently, the final feature representation is constructed using only the "consensus" of reliable, aligned views.
>
> **New Experiments (Table 3 & Figure 9):**
> To quantify this, we introduced a controlled experiment where we artificially corrupted a subset of training poses (adding $1^{\circ}$ rotational and 1% scene-scale translational noise) while keeping the geometry fixed.
>
> **Quantitative Results (from Table 3):**
> As shown below, our method maintains state-of-the-art performance even under severe noise. For example, with **20% of camera poses corrupted**, the mean mIoU remains at **65.0%**, which is statistically identical to the noise-free baseline (**65.1%**).
>
> | Corrupted Images (per 100) | Figurines | Ramen | Teatime | Waldo Kitchen | Mean |
> | :--- | :--- | :--- | :--- | :--- | :--- |
> | 100/100 | 1.43 | 56.0 | 18.1 | 36.8 | 28.1 |
> | 50/100 | 64.7 | **71.7** | 58.9 | 48.1 | 60.8 |
> | 20/100 | **72.1** | 63.3 | **69.1** | 55.2 | 65.0 |
> | 10/100 | 68.8 | 64.8 | 63.9 | 57.4 | 63.7 |
> | 3/100 | 65.6 | 64.2 | 63.4 | 59.5 | 63.2 |
> | 2/100 | 66.9 | 64.2 | 63.4 | 59.1 | 63.4 |
> | 1/100 | 66.2 | 63.0 | 64.7 | 59.8 | 63.6 |
> | **0/100 (Clean)** | 67.6 | 68.5 | 62.3 | **62.1** | **65.1** |
>
> **Qualitative Results:**
> **Figure 9** visually demonstrates that even with 50% of images corrupted, the segmentation masks remain coherent and accurate.
>
> ### **2. Integration of Segmentation Cues (SAM)**
> **Summary:** We agree that segmentation cues are valuable. In fact, we *do* incorporate SAM-based cues, utilizing them in the **Post-Lifting Aggregation** stage to stabilize feature lifting in ambiguous regions.
>
> **Integration Strategy:**
> The reviewer asked if soft grouping signals are at odds with our analytical framework. They are not; however, rather than introducing them as soft weights *during* the linear solve (which might complicate the closed-form solution), we utilize them as a validation prior *after* the initial lift.
> * By comparing the initial lifted features against SAM masks (as described in the mechanism above), we effectively use segmentation cues to resolve ambiguities.
> * This approach allows us to retain the efficiency of the closed-form solver while still leveraging the semantic grouping power of foundation models like SAM to handle low-texture or ambiguous regions.
>
> We hope these additional experiments and clarifications address your concerns regarding the robustness and flexibility of our pipeline.

---

### Official Review · Reviewer_u6Yd · 2025-11-07

**Soundness:** 3
**Presentation:** 2
**Contribution:** 3
**Rating:** 6
**Confidence:** 3

**Summary:**

This paper addresses the feature lifting problem, which aims to transform vision foundation model features (e.g., DINO/CLIP) into Gaussian-splatting-based representations. The authors innovatively formulate this problem as a sparse linear inverse problem and propose a closed-form solver. In addition, they introduce two modules to suppress the noise inherent in foundation model features. Experimental results demonstrate the proposed method’s effectiveness and generalization across different Gaussian splatting kernels. Overall, the paper presents solid contributions both theoretically and practically.

**Strengths:**

* The paper formulates feature lifting as a sparse linear inverse problem and derives a closed-form solution, which is elegant and theoretically sound.
* The mathematical derivations and reasoning are solid and well-motivated.

**Weaknesses:**

* The space allocation in the manuscript is unbalanced — too few visualizations are included in the main text, while most figures are deferred to the supplementary material. Moreover, some figure captions are vague.
* The paper lacks discussion and comparison with feed-forward models related to VGGT, such as Anysplat, which can also lift DINOv2 features to Gaussian-splatting representations. Considering their feed-forward nature, such models are likely to offer faster runtime performance.
* In Table 1 (b) and (c) lack highlighted numerical values, making it difficult to visually discern the performance trends.
* The statement “Third, many existing methods are specialized for particular feature types or geometric kernels, which may limit generalization across broader settings” requires further clarification. Specifically, which feature types are those methods specialized for? It would also strengthen the paper to include visual evidence showing that the proposed approach generalizes better across diverse cases.

**Questions:**

* In Figure 2, why are the segmentation boundaries so noisy? Moreover, for the two similar and adjacent eggs, why does the method only able to segment one of them?
* According to Section 3.2, the solver is closed-form and enables one-shot estimation without iterative SGD. Theoretically, this should result in very fast inference, yet the runtime is still reported as 1–3 minutes. Could the authors clarify what factors contribute to this computational cost?

---

> ### Author Response · Authors · 2025-11-17
> **Justification to Weakness and Response to Questions**
>
> # Response to Reviewer U6YD
>
> We thank the reviewer for their constructive feedback.
>
> ***Action Taken***: All changes in the revised PDF are highlighted in blue.
>
> We invite the reviewer to explore the supplementary ZIP, which contains multiple  ***interactive demos*** (via `index.html` in the "anonymous website" folder) for a more detailed exploration of our results. Please unzip whole folder and click the `index.html` to visualize
>
> We address the specific concerns below:
>
> ---
>
> ### **• On Weakness I (Visualizations)**
>
> - ***Action Taken***: We have revised the manuscript to move Figure 7 (Attention Map Comparison) from the appendix to the main paper to better illustrate the attention quality. We have updated Figure 2 for font consistency and expanded the caption of Figure 3 to provide deeper context on the qualitative comparisons.
>
> - We are actively revising the layout to bring more key visualizations into the main text to better balance the theoretical and visual content.
>
>
>
> ---
>
> ### **• On Weakness II (Anysplat/VGGT)**
> We appreciate the suggestion. We re-examined the Anysplat and VGGT papers and ***could not*** find an official implementation or description of their method being used for lifting DINOv2 or other dense semantic features. Our understanding is that these methods focus on geometry and appearance, not on lifting arbitrary semantic features. If we have overlooked a specific variant or section, we would be grateful for the pointer and will gladly incorporate the comparison.
>
> ---
>
> ### **• On Weakness III (Table 1b/c)**
> - ***Action Taken***: We appreciate the suggestion. We added highlighting to Table 1(c) in this revision.
>
> - For Table 1(b), we respectfully clarify that its purpose is ***not*** to show a single “best” result, but to demonstrate **robustness** across diverse feature types (as discussed in lines 432–441). The table shows that our method ***consistently*** achieves high cosine similarity (80%–96%) for all features (CLIP, DINO, ResNet, etc.), validating its feature-agnostic nature. Highlighting a single value would misrepresent this core contribution.
>
> ---
>
> ### **• On Weakness IV (Generalization)**
>
> - Our statement refers to the fact that prior works (detailed in our related work) ***exclusively*** focus on the “SAM + OpenCLIP” feature and “3DGS” kernel combination. Each type of approaches ***often face specific limitations***: grouping-based methods are constrained by their need for segmentation masks (like those from SAM), and heuristic methods lacked a theoretical basis for generalization. Training-based methods has extensively training time. Our work provides this theoretical grounding, enabling generalization beyond these specific while been much more efficient.
>
> - Visual evidence of this generalization is provided in Figures 3, 4, and 5, and in the interactive supplementary website.
>
> - ***Action Taken***: We clarify our statement in the paper for in line 75 to 76
> ---
>
> ### **• On Question I (Noisy Eggs)**
> The noisy boundary is an artifact of the **hard thresholding** step used to generate the final segmentation mask, not an issue with the underlying feature. As shown in Figure 6 (“Our Attention Maps”), the raw attention map for “egg” is smooth and correctly identifies **both** eggs. The specific hard threshold chosen for this visualization simply fails to capture the second egg. To demonstrate that the underlying feature map is correct, the raw attention map can be viewed here:
>
> **[egg attention map (IMG)](https://imagedelivery.net/qlt0ZlAaeYqb8ayCDU7vpA/a58b59c2-4c70-47d8-1a27-c42496b13700/public)**
>
> ---
>
> ### **• On Question II (Runtime)**
> The 1–3 minute runtime is not due to iterative optimization (which we avoid), but to the I/O and computation required for the one-shot solution on high-dimensional data. The runtime scales linearly with the number of images, their resolution (\(H, W\)), and the feature dimension (\(F\)).
> The time per frame can be modeled as:
>
> $$
> T_{\text{frame}} = (a + cF) \times H \times W
> $$
>
> Where \(a\) is fixed overhead (e.g., depth sorting) and \(c\) is a small constant for the blending operation.
> When \(F = 3\) (RGB), the time is approximately \(0.03\)s/frame, enabling real-time performance.
> For \(F = 512\), this increases to approximately \(0.6\)s/frame.
> For a 300-frame scene (e.g., Figurines), this results in ~3 minutes.
>
> To empirically show this linear scaling, we provide a benchmark on a 670-image scene (448×448 resolution) on an RTX 4090:
>
> | Channel Count (#F) | Total Time (sec) |
> |--------------------|------------------|
> | 32                 | 20               |
> | 64                 | 31               |
> | 128                | 68               |
> | 256                | 127              |
> | 512                | 245              |
>
> This remains significantly faster than comparable methods, which can often require over an hour. For tasks like style transfer (lifting 3-channel features), our method achieves ***real-time*** runtimes.
>
> ---

---

> > ### Author Response · Authors · 2025-11-25
> > **Additional Information Regarding Table 1(b)**
> >
> > **Action Taken**: We find that adding more comparison regarding Tab.1(b) will bring more context and help understand our contribution. Therefore, we add one additional section in appendix H for feature agnostisim.
> >
> > **Comparative Results (Mean Cosine Similarity)**
> > We conducted a new experiment on the LeRF dataset. As shown in the consolidated table below (with a comprehensive breakdown added to **Table 8 in the Appendix**), our method consistently outperforms both baselines across every feature type tested. We also added ***Fig.12 and Fig.13*** for further qualitative comparison.
> >
> > | Feature Type | Semantic Gaussian | Argmax Lifting | Ours | Improvement |
> > | :--- | :---: | :---: | :---: | :---: |
> > | **SAM+OpenCLIP** | 83.2 | 83.4 | **90.4** | **+7.0** |
> > | **MaskCLIP** | 92.8 | 92.8 | **93.9** | **+1.1** |
> > | **CLIP** | 92.8 | 92.5 | **93.8** | **+1.0** |
> > | **DINO** | 76.4 | 75.9 | **81.0** | **+4.6** |
> > | **DINOv2** | 83.0 | 83.6 | **87.0** | **+3.4** |
> > | **ViT** | 83.9 | 82.3 | **85.7** | **+1.8** |
> > | **ResNet** | 94.9 | 94.9 | **96.0** | **+1.1** |
> >
> >
> > This empirical superiority directly validates our theoretical proof (Lines 310–314):
> > * **Baselines:** Heuristic methods (Argmax) and standard optimizations (Semantic Gaussian) often fail to account for the sparse, row-stochastic nature of the lifting problem or get trapped in local optima, leading to higher reconstruction errors.
> > * **Ours:** By deriving a closed-form solver with a provable upper bound on the global optimal error, our method mathematically minimizes deviation from the "true" projected feature. This results in consistently higher fidelity lifting regardless of the underlying feature architecture.
> >
> > We have updated the manuscript to include this comparative analysis, ensuring readers can properly contextualize the strength of our feature-agnostic performance. We are also actively considering would it be possible to move this part of description to the main paper.

---

> > ### Comment · Reviewer_u6Yd · 2025-11-27
> > **Response to Authors**
> >
> > Thank you for the clear and detailed reply. My concerns have been fully addressed, and I will maintain my rating of 6.

---

### Author Response · Authors · 2025-11-21
**Response to All Reviewer**

# General Response to All Reviewers

We sincerely thank all reviewers for their time, valuable feedback, and constructive suggestions. Based on the feedback, we have carefully revised the manuscript to improve clarity, formatting, and visualization. Below is a summary of the general updates:

* **Visualizations:** We have reorganized the figures to bring key qualitative comparisons (previously in the appendix) into the main text. We have also updated font sizes and styles in Figures 1, 2, 4, 5, and 6 for consistency.
* **Interactive Demos:** We invite reviewers to explore the supplementary ZIP file. Please unzip the folder and open `index.html` in the "anonymous website" folder. This contains interactive demos that allow for a detailed, 3D exploration of our results, which complements the static figures in the paper.
* **Manuscript Updates:** All modifications in the updated PDF are marked in ***BLUE***  for easy tracking.

---

### Meta-Review · Area_Chair_FgfA · 2026-01-06

**Summary:**

The paper received three reviews initially; however, one review was flawed.  Despite this flaw, the reviewer provided additional comments after the rebuttal, and the authors responded to those comments.

The primary concerns are the experimental results presented in the tables, the visualizations, the qualitative outcomes, the formatting, and the run-time. The rebuttal and discussions have sufficiently addressed these issues. The revised PDF contains additional experimental results and visualizations; however, the area chair still notes several punctuation errors, cluttered presentations of equations, an inconsistent citation style, and redundant spaces throughout the manuscript. The authors are encouraged to proofread the paper more carefully. Overall, the area chair recommends accepting the paper.

**Reviewer Concerns:**

The reviewers' main concerns were addressed, but the formatting and writing could be improved further.

**Reviewer Scores:**

On 27 Nov 2025, **Reviewer u6Yd** stated
> My concerns have been fully addressed, and I will maintain my rating of 6.

---

**Reviewer 67o8** did not indicate whether the score would be revised.
The initial review contained flaws, so the comments were not taken into account in the final decision.

The reviewer's follow-up questions about camera pose noise and segmentation cues were more relevant. The authors submitted additional experimental results to address the camera pose issue. They also clarified how segmentation cues were utilized during the post-lifting aggregation stage.

The area chair expects that the reviewer will raise the score based on these clarifications.

---

On 25 Nov 2025, **Reviewer djJk** mentioned
> I believe the second weakness has been addressed.

However, the reviewer still had concerns about other issues and noted
> Given the above issues, I maintain my rating of 4.

The authors later provided follow-up responses to address the issues related to Table 1b and formatting, but the reviewer did not reply.
The area chair has reviewed the comments and responses, along with the revised PDF, and believes that the concerns have been adequately addressed. The area chair expects **Reviewer djJk** to raise the score.

---

The final expected scores are **6**, **6**, and **6**.

---

### Decision · Program_Chairs · 2026-01-26

Accept (Poster)